# Deep Learning on a Data Diet: Finding Important Examples Early in Training

**Mansheej Paul**
Stanford University
mansheej@stanford.edu

**Surya Ganguli**
Stanford University; Facebook AI Research
sganguli@stanford.edu

**Gintare Karolina Dziugaite**
Mila *
gkdz@google.com

## Abstract

Recent success in deep learning has partially been driven by training increasingly overparametrized networks on ever larger datasets. It is therefore natural to ask: how much of the data is superfluous, which examples are important for generalization, and how do we find them? In this work, we make the striking observation that, in standard vision datasets, simple scores averaged over several weight initializations can be used to identify important examples *very early in training*. We propose two such scores—the Gradient Normed (GraNd) and the Error L2-Norm (EL2N) scores—and demonstrate their efficacy on a range of architectures and datasets by pruning significant fractions of training data without sacrificing test accuracy. In fact, using EL2N scores calculated a few epochs into training, we can prune half of the CIFAR10 training set while slightly improving test accuracy. Furthermore, for a given dataset, EL2N scores from one architecture or hyperparameter configuration generalize to other configurations. Compared to recent work that prunes data by discarding examples that are rarely forgotten *over the course of training*, our scores use only *local information early in training*. We also use our scores to detect noisy examples and study training dynamics through the lens of important examples—we investigate how the data distribution shapes the loss surface and identify subspaces of the model's data representation that are relatively stable over training.

## 1 Introduction

Recently, deep learning has made remarkable progress driven, in part, by training overparameterized models on ever larger datasets. This trend creates new challenges: the large computational resources required pose a roadblock to the democratization of AI. Memory and resource constrained settings, such as on-device computing, require smaller models and datasets. Identifying important training data plays a role in online and active learning. Finally, it is of theoretical interest to understand how individual examples and sub-populations of training examples influence learning.

To address these challenges, we propose a scoring method that can be used to identify important and difficult examples early in training, and prune the training dataset without large sacrifices in test accuracy. We also investigate how different sub-populations of the training data identified by our score affect the loss surface and training dynamics of the model.

---

* This work was carried out while the author was at ServiceNow. It was finalized at Google Brain.

35th Conference on Neural Information Processing Systems (NeurIPS 2021).

Recent work on pruning data [1, 2], can be placed in the broader context of identifying coresets—examples that provably guarantee a small gap in training error on the full dataset [3–7]. However, due to the nonconvex nature of deep learning, coreset techniques make conservative estimates that lead to weak theoretical guarantees and are less effective in practice.

A different approach recently proposed by Toneva et al. [8] tracks the number of times through training an example transitions from being correctly classified to misclassified, called a "forgetting event", and find that some examples are rarely forgotten, while others are forgotten repeatedly. Empirically, they observed that training accuracy is not affected by the rarely forgotten training examples and a large fraction of the training data can be removed without any impact on test accuracy. However, since this method relies on collecting forgetting statistics throughout training, the forgetting score is typically calculated in the middle of or at the end of training. Toneva et al. [8] find that, in their example of a ResNet18 trained on CIFAR-10 for 200 epochs, the Spearman rank correlation between early and late scores is good after about 25 epochs and stabilizes after 75 epochs.

Broadly speaking, the ability to prune datasets raises a number of questions: What is the nature of examples that can be removed from the training data without hurting accuracy? How early in training can we recognize such examples? How many examples do we need and how does this depend on the data distribution? These questions may have no generic answers and so, in this work, we begin to pursue them empirically in the context of several standard vision benchmarks and standard network architectures. Answers to these questions may both (1) lead to new methodologies that could dramatically reduce training times and memory requirements, and (2) offer important insights into the training dynamics of deep neural networks, and the role of data.

Our first finding is that *very early in training* (just a few epochs), partial forgetting scores identify large fractions of data that can be pruned. Analyzing this puzzling result with a one gradient step analysis of training suggests a very simple heuristic: use the loss gradient norm of individual examples to identify important examples. While this approach does not work when the loss gradient norms are computed at the weights early in training of a single trajectory, we find that, surprisingly, averaging these norms over multiple weight initializations does produce a ranking that correlates strongly with forgetting scores and allows us to prune a significant fraction of examples early in training. Indeed, we can prune 50% of examples from CIFAR-10 without affecting accuracy, while on the more challenging CIFAR-100 dataset, we can prune 25% of examples with only a 1% drop in accuracy.

Through a series of empirical studies, we have begun to tease apart the properties of important examples and how they can depend on the data distribution. In particular, we find that the examples with the very highest norms become superfluous as the amount of label noise increases. Indeed, even on clean data, we find that in the high pruning regime, the best population excludes the very highest-scoring examples.

## 1.1 Contributions

- We propose to score the importance of each training example $(x_i, y_i)$ by its expected loss gradient norm (GraNd score), which, up to a constant, bounds the expected change in loss for an arbitrary example $(x, y)$ caused by removing $(x_i, y_i)$.

- We show that pruning training samples with small GraNd scores at initialization allows one to train on much smaller subset of the training data without significant loss in accuracy. While the pruning levels are comparable to those provided by other methods [1, 8], our score is the only one that is well-defined at initialization and early in training.

- Our experimental findings suggest that, within the first few epochs of training, the GraNd score is well-approximated by the norm of the error vector (EL2N score), where the error vector is the predicted class probabilities minus one-hot label encoding. In fact, we find that the EL2N score provides an even stronger signal for data-pruning—for CIFAR10 we can prune 50% of the data, and for the harder CIFAR100, we can prune as much as 25% of the data without any loss in test accuracy.

- We study the role of examples with the highest EL2N scores, and find that excluding a small subset of the very highest scoring examples produces a boost in performance. This boost in performance is enhanced in a corrupted label regime.

- We introduce a method, based on linearly connected modes, for studying the empirical risk surface in terms of the modes of *subsets of data*, allowing us to identify when, in training, the final performance on subpopulations is determined. We demonstrate that the linearly connected mode at-convergence of empirical risk surface computed on low EL2N score examples is determined much earlier in training compared to high score examples.

- Finally, we study how an example's EL2N score connects to the network's training dynamics. We do so by tracking the data-dependent NTK submatrices corresponding to the low or high score examples, and measuring the rate at which it evolves in a scale-invariant way. We find that the NTK submatrix for the high score examples evolves faster throughout training, supporting our hypothesis that high-scoring examples are the ones driving the learning and the changes in the NTK feature space [9].

## 2 Which samples are important for learning?

### 2.1 Preliminaries

We consider supervised classification, where $S = \{(x_i, y_i)\}_{i=1}^N$ denotes the training set, drawn i.i.d. from an unknown data distribution $\mathcal{D}$, with input vectors $x \in \mathbb{R}^d$ and one-hot vectors $y \in \{0,1\}^K$ encoding labels. For a fixed neural network architecture, let $f_{\mathbf{w}}(x) \in \mathbb{R}^K$ be the logit outputs of the neural network with weights $\mathbf{w} \in \mathcal{W} \subseteq \mathbb{R}^D$ on input $x \in \mathbb{R}^d$. Let $\sigma$ be the softmax function given by $\sigma(z_1, \ldots, z_K)_k = \exp\{z_k\} / \sum_{k'=1}^K \exp\{z_{k'}\}$. Let $p(\mathbf{w}, x) = \sigma(f(\mathbf{w}, x))$ denote the neural network output in the form of a probability vector. For any probability vector $\hat{p}$, let $\ell(\hat{p}, y) = \sum_{k=1}^K y^{(k)} \log \hat{p}^{(k)}$ denote cross-entropy loss.

Let $\mathbf{w}_0, \mathbf{w}_1, \mathbf{w}_2, \ldots, \mathbf{w}_T$ be the iterates of stochastic gradient descent (SGD), where, for some sequence of minibatches $S_0, S_1, \ldots, S_{T-1} \subseteq S$ of size $M$, we have

$$\mathbf{w}_t = \mathbf{w}_{t-1} - \eta \sum_{(x,y) \in S_{t-1}} g_{t-1}(x, y), \tag{1}$$

for $g_{t-1}(x, y) = \nabla_{\mathbf{w}_{t-1}} \ell(p(\mathbf{w}_{t-1}, x), y)$, and $t = 1, \ldots, T$.

### 2.2 Gradient Norm Score and an infinitesimal analysis

Fix a training set $S$. Due to training with SGD from a random initialization, the weight vector at time $t > 0$, $\mathbf{w}_t$, is a random variable. The expected magnitude of the loss vector is our primary focus:

**Definition 2.1.** The GraNd score of a training example $(x, y)$ at time $t$ is $\chi_t(x, y) = \mathbb{E}_{\mathbf{w}_t} \|g_t(x, y)\|_2$.

Here we describe conditions under which the GraNd score controls the contribution of a training example to the change in the training loss. In order to simplify our analysis, we approximate the training dynamics as if they were in continuous time.

A key quantity in our analysis is the time derivative of the loss for a generic labeled example $(x, y)$: $\Delta_t((x, y), S_t) = -\frac{d\ell(f_t(x), y)}{dt}$ (where $f_t(\cdot) = f_{\mathbf{w}_t}(\cdot)$), i.e., the instantaneous rate of change in the loss on $(x, y)$ at time $t$, where the gradient is computed on the minibatch $S_t$. By the chain rule,

$$\Delta_t((x, y), S_t) = g_t(x, y) \frac{d\mathbf{w}_t}{dt}. \tag{2}$$

This relates to our discrete time dynamics via $\frac{d\mathbf{w}_t}{dt} \approx \mathbf{w}_{t+1} - \mathbf{w}_t = -\eta \sum_{(x',y') \in S_t} g_t(x', y')$.

Our goal is to understand how removing a training point from minibatch $S_t$ affects $\Delta_t((x^*, y^*), S_t)$ for any $(x^*, y^*)$. If a training point $(x, y)$ is not in the minibatch $S_t$, then the effect is trivial. We thus study $\Delta_t((x^*, y^*), S)$ in order to be able to rank all the training examples.

**Lemma 2.2.** Let $S_{\neg j} = S \setminus (x_j, y_j)$. Then for all $(x^*, y^*)$, there exists $c$ such that

$$\|\Delta_t((x^*, y^*), S) - \Delta_t((x^*, y^*), S_{\neg j})\| \leq c \|g_t(x_j, y_j)\|. \tag{3}$$

*Proof.* For a given example $(x^*, y^*)$, the chain rule yields $\Delta_t((x^*, y^*), S) = -\frac{d\ell(f_t(x^*), y^*)}{dt} = \frac{d\ell(f_t(x^*), y^*)}{d\mathbf{w}_t} \frac{d\mathbf{w}_t}{dt}$. Since the weights are updated using SGD, we have $\frac{d\mathbf{w}_t}{dt} = -\eta \sum_{(x_j, y_j) \in S_t} g_t(x_j, y_j)$. Letting $c = \eta \|\frac{d\ell(f_t(x^*), y^*)}{d\mathbf{w}_t}\|$, the result follows. $\square$

At any given training step, given the current location $\mathbf{w}_t$, the contribution of a training example $(x, y)$[2] to the decrease of loss on any other example, is bounded by Eq. (3). Since the constant $c$ does not depend on the training example $(x, y)$[3], we only consider the gradient norm term, $\|g_t(x, y)\|$. The expected value of this gradient norm is exactly the GraNd score of $(x, y)$. In other words, examples with a small GraNd score in expectation have a bounded influence on learning how to classify the rest of the training data at a given training time[4]. We therefore propose to rank training examples by their GraNd scores, larger norm meaning more important for preserving $\Delta_t(x)$.

For an arbitrary input $x \in \mathbb{R}^d$, let $\psi_t^{(k)}(x) = \nabla_{\mathbf{w}_t} f_t^{(k)}(x)$ denote the $k$th logit gradient. Then GraNd can be written as[5]

$$\chi_t(x, y) = \mathbb{E} \left\| \sum_{k=1}^{K} \nabla_{f^{(k)}} \ell(f_t(x), y)^T \psi_t^{(k)}(x) \right\|_2. \tag{4}$$

Under the cross entropy loss, $\nabla_{f^{(k)}} \ell(f_t(x), y)^T = p(\mathbf{w}_t, x)^{(k)} - y_k$. When $\{\psi_t^{(k)}(x)\}_k$ are roughly orthogonal across logits, and are of a similar size across logits and training examples $x$, then we can approximate GraNd by just the norm of the error vector.

**Definition 2.3.** The EL2N score of a training sample $(x, y)$ is defined to be $\mathbb{E}\|p(\mathbf{w}_t, x) - y\|_2$.

Our experimental results suggest that this approximation becomes accurate after a few epochs of training (see Section 3). These approximations are also in agreement with the empirical results reported in [9, 10]. Fort and Ganguli [10, Sec. 5.1] demonstrate that the mean logit gradients are nearly orthogonal among classes throughout training. The authors demonstrate that per-example gradients cluster around the mean logit gradient. Fort et al. [9, Figs. 12D-14D] provide evidence that the mean logit gradient directions evolve rapidly early in training and then stabilize (as measured by the cosine distance between mean logit gradient vectors at different times in training).

## 2.3 Comparison to forgetting scores

Toneva et al. [8] define a "forgetting event" for a training sample to be a point in training when the classifier switches from making a correct classification decision to an incorrect one. They define an approximate *forgetting score* for each training example as the number of times during training when it was included in a minibatch *and* underwent a forgetting event. Toneva et al. demonstrate that examples with low forgetting score may be completely omitted during training without any noticeable effect on the accuracy of the learned predictor. In Fig. 1 and Appendix E.5, we make an empirical comparison of forgetting scores to our proposed GraNd and EL2N scores.

In Lemma 2.2, we bounded the contribution of a training example to the decrease of the loss of any other sample over a single gradient step. Due to $\psi_t(\cdot)$'s being time-dependent, it is complicated to extend the analysis to multiple steps. However, it is interesting to consider a case when $\psi_t(x_i) = \psi(x_i)$ for all $x_i$ in the training set, and $K = 1$. Then summing the bound in Eq. (3) on how much a sample $(x_j, y_j)$ affects the logit output on an arbitrary point at each time $t \in \{1, .., T\}$, we obtain a score that depends on $\|\psi(x_j)\| \sum_t (p_t(x_j) - y_j)|$. For two examples, $(x, y)$ and $(x', y')$, such that $\|\psi(x')\| \approx \|\psi(x)\|$, we see that the example that is learned faster and maintains small error over training time will have a smaller GraNd score on average throughout training. Note that $|(p_t(x_j) - y_j)|$, if rescaled, is an upper bound on 0–1 loss, and therefore $\sum_t |(p_t(x_j) - y_j)|$ upper bounds the number of forgetting events during training (after rescaling). In this simplified setting an example with a high number of forgetting events will also have a high GraNd score.

# 3 Empirical Evaluation of GraNd and EL2N Scores via Data Pruning

In the previous section, we motivated GraNd and EL2N scores by quantifying the influence of a training example on the loss of an arbitrary example after one optimization step. In this section,

---

[2]Here we drop the index $j$ since we refer to an arbitrary training point.

[3]Note that $c$ depends on $(x^*, y^*)$ but for a given $(x^*, y^*)$, $c$ is fixed for all training inputs $(x, y)$, allowing us to rank the training examples.

[4]The opposite is not necessarily true: examples with large scores may have gradients that cancel out and do not contribute much, meaning that this upper bound is loose.

[5]The score $\chi_t(x, y)$ is a function of $(x, y)$. Thus $(x, y)$ is non-random, and the expectation is taken over the remaining randomness (the weights at time t which depend on a random initialization, random minibatch sequence, GPU noise, etc.).

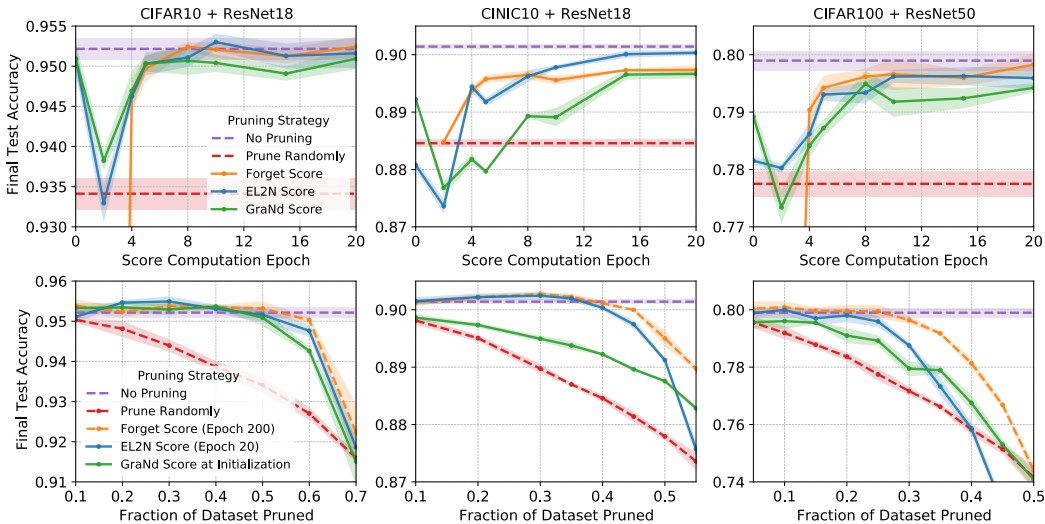

Figure 1: Columns correspond to three different dataset and network combinations (labeled at the top). Each legend applies to all 3 figures in its row. *First row:* Final test accuracy achieved by training on a subset of training data comprised of examples with maximum forgetting, EL2N and GraNd scores computed at different times early in training. Subsets of a fixed size are used: networks are trained on 50% of training data for CIFAR-10, 60% for CINIC-10 and 75% for CIFAR-100. *Second row:* Final test accuracy achieved by training after different fractions of the dataset are pruned. Here we compare forgetting scores at the end of training, EL2N scores early in training (at epoch 20) and GraNd scores at initialization. In each case, examples with the lowest scores are pruned at initialization. *In all experiments* accuracies achieved by training on the full dataset and on a random subset of the corresponding size are used as baselines.

we evaluate these scores empirically, and verify that they identify examples important for generalization. Networks trained on subsets of the data with high scores achieve levels of test accuracy comparable to training on the full dataset and are competitive with other state of the art data pruning methods. Perhaps most remarkably, these scores are effective even when computed early in training and perform significantly better than a random baseline, even at initialization.

**Data pruning experiments.** We train convolutional neural networks of varying depth–ResNet18 and ResNet50 [11]–on standard vision datasets of varying difficulty–CIFAR-10, CIFAR-100 [12], and CINIC-10 [13]. All scores are calculated by averaging the scores from ten independent training runs. After calculating scores and selecting a training subset, final test accuracies are obtained by retraining networks from new random initializations on only the selected subset. Networks used for evaluating the scores are initialized with seeds that are different from those used to calculate the scores. For each experiment, we report the mean of four independent runs and represent variability across runs by shading the region which spans the 16th to 84th percentile of obtained accuracies. See Appendix B for more implementation details and Appendix E for additional experiments.

In Fig. 1, we show the results of two sets of experiments (top and bottom) on three different network and dataset combinations. The first experiment asks, how early in training are forgetting, GraNd and EL2N scores effective at identifying examples important for generalization? We compare the final test accuracy from training on subsets of fixed size but pruned based on scores computed at different times early in training. The second experiment compares how GraNd scores at initialization, EL2N scores early in training and forgetting scores at the end of training negotiate the trade-off between generalization performance and training set size. The training sets are constructed by pruning different fractions of the lowest score examples. In all examples, training on the full dataset and a random subset of the corresponding size are used as baselines. We make the following observations.

**Pruning at initialization.** In all settings, GraNd scores can be used to select a training subset *at initialization* that achieves test accuracy significantly better than random, and in some cases, competitive with training on all the data. This is remarkable because GraNd only contains information about the gradient norm at initializion, averaged over initializations. This suggests that the ge-

ometry of the training distribution induced by a random network contains a surprising amount of information about the structure of the classification problem. EL2N scores, which only contain information about errors, are not consistently effective at initialization and forgetting scores, which require counting forgetting events over training, are not defined at initialization.

**Pruning early in training.** We find that, after only a few epochs of training, EL2N scores are extremely effective at identifying important examples for generalization. For a wide range of intermediate pruning levels, training on the highest scores performs on par with or better than training on the full dataset. Even at higher pruning levels, EL2N scores computed using local information early in training are competitive with forgetting scores which integrate information over the training trajectory. This suggests that the average error vector *a few epochs into training* can identify examples that the network heavily uses to shape the decision boundary *throughout training*.

Interestingly, at extreme levels of pruning with either EL2N or GraNd scores, we observe a sharp drop in performance. We hypothesize that this is because at high levels of pruning, using either GraNd or EL2N scores leads to bad coverage of the data distribution. By only focusing on the highest error examples, it is likely that an entire subpopulation of significant size that is present in the test data is now excluded from the training set. We only fit a small number of very difficult examples and do not keep enough of a variety of examples for training models with good test error.

**A property of the data.** Our results suggest that the ranking of important examples induced by EL2N scores is a property of the dataset and not specific to a network. First, in Appendix E.2, we show that a ResNet18 and a ResNet50 trained on CIFAR-10 have similar performance curves and the same amount of data can be pruned, even though ResNet50 is a much deeper network with more parameters. Second, EL2N scores calculated on one set of network architecture and hyperparameter configurations can be used to prune data for training with a different network architecture or hyperparameter configuration. The set of important examples generalizes across architectures and hyperparameters. See Appendix E.3 for the experiment on generalization across architectures and Appendix E.4 for the experiment on using scores calculated during hyperparameter optimization. Additionally, in an analysis of the sensitivity of the scoring methods to hyperparameters in Appendix E.1, we observe that scores calculated on a single network do not perform as well as those averaged across networks.

We hypothesize that averaging the gradient or error norms over multiple initializations or training trajectories removes dependence on specific weights, allowing a more accurate distillation of the properties of the dataset. EL2N scores can thus be used to probe and understand how the distribution of the training data impacts dynamics (as we show in the next sections). Additionally, it can also reduce the computational burden of training neural networks; once we compute the scores, future networks can be trained on the pruned dataset.

In the following experiments, we focus on EL2N scores computed early in training, as they appear to more accurately identify important examples.

## 4   Identifying noise examples

In the previous section, we studied the effect of keeping the highest-scoring examples, and found that we could train on only the top 50% of examples by score without a drop in accuracy (CIFAR-10). What is the nature of subpopulations of examples that allow us to reach high accuracy? One hypothesis is that the highest-scoring examples are the most important ones for achieving an accurate classifier. In this section, we refute this hypothesis, and demonstrate the role of label noise.

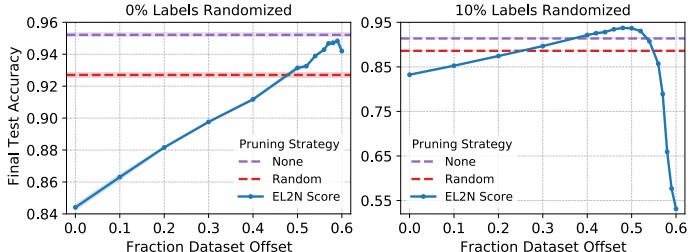

Figure 2: ResNet18 trained on a 40% subset of CIFAR-10 with clean *(left)* and 10% randomized labels *(right)*. The training subset contains the *lowest* scoring examples *after* examples with scores below the offset are discarded. Scores computed at epoch 10.

To test whether the highest-scoring examples are most important for achieving high accuracy, we first sort the examples by increasing EL2N score computed after a small number of training epochs.[6] Then we perform a sliding window analysis by training on a subset of examples with scores within a window from percentile $f$ to percentile $f + P$ percentile, always keep $P\%$ of the data but sliding up $f$. As this window slides to higher percentiles, performance increases, except when the window includes examples with the very highest scores Fig. 2 (left). Indeed the the optimal sliding window actually excludes approximately $500$ of the highest-scoring training examples. These effects are reduced in the low pruning regime (see Appendix F.1). In Appendix C, we visualize some of the images that are excluded from each class.

Before we analyze these results, we first place them into a wider context, where we also change the amount of noise in the underlying label distribution. We repeat the experiment outlined above, but corrupt a random $K\%$ of labels, replacing them with a random label, mirroring the protocol popularized by Zhang et al. [14]. Fig. 2 reveals that with increased label corruption, the optimal window shifts and excludes a higher number of examples. Therefore, the effect we see in the noiseless case appears to be magnified in the presence of label noise. Appendix F.2 examines how adding label noise influences the distribution of EL2N scores of examples.

These findings have several implications. The most obvious implication is that training with only the highest-scoring samples may not be optimal, especially when there is label noise. When the population has a low Bayes error rate, using only the highest scoring samples yields optimal results. However, without a validation set, one should be cautious in excluding high-score examples. Feldman [15] discusses memorization in a noisy-label setup and gives conditions under which one should memorize in order to not misclassify singleton examples ( examples in the training data that are the sole representatives of a subpopulation). For example, if the subpopulation appears with a frequency $\Omega(1/N)$, memorizing such examples can improve generalization. In practice, we may not know whether our data fits these conditions. However, our analysis in Fig. 2 suggests a simple and powerful method to prune data for optimal performance by optimizing just two hyperparameters of a sliding window using a validation set.

## 5 Optimization landscape and the training dynamics

### 5.1 Evolution of the data-dependent NTK

The dynamics of neural-network training in the infinite-width limit are now well understood [16, 17]: for an appropriate scaling of the learning rate and initial weights, the neural network behaves like a linear model in which the data is transformed by the Neural Tangent Kernel (NTK) at initialization, which is defined as the product of the Jacobians of the logits at initialization. In the limit, neural network training implements kernel regression with the fixed NTK as the kernel.

However, finite neural networks outperform their infinite-width limits [18] and have different dynamics early in training [19]. In fact, rather than being constant, the data-dependent NTK, defined by Fort et al. [9] as the Gram matrix of the logit Jacobian, evolves with high velocity in the initial phase of training. Then, around the time of onset of linear mode connectivity, the NTK velocity stabilizes at a smaller value and remains nearly constant for the rest of the high learning rate training time.

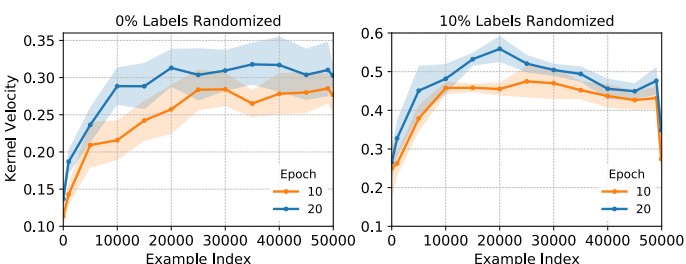

Figure 3: Kernel velocity for different subsets of images when ResNet18 is trained on CIFAR-10 with all true labels *(left)* and 10% label noise *(right)*. Examples are sorted in ascending order by EL2N scores and each point corresponds to the kernel velocity of 100 contiguous images starting at example index. Both scores and velocities are computed at the same epoch indicated by color.

---

[6]In Appendix F.3, we repeat these experiments for the GraNd score.

Here we seek to understand which training samples contribute to the NTK gram matrix evolution. To empirically approximate the velocity of a NTK submatrix corresponding to a subset of images in a scale invariant way, we follow [9]. We compute the cosine distance between two NTK gram matrices on the given subset, one computed at epoch $t$, and another one at epoch $t+1$, one epoch later (see Appendix B.3). We look at submatrices of a fixed size, formed by examples with contiguous EL2N scores. Fig. 3 shows that higher EL2N scores lead to higher velocities. This relationship is not affected by the time at which both are computed.

Interestingly, the kernel velocity drops off sharply for examples with the very highest scores when label noise is introduced. In Section 4, we showed that dropping these examples boosts the accuracy of the final predictor. We hypothesize that, while the kernel velocity is higher for harder examples that the model is actively trying to fit, the kernel velocity drops off for the very highest scoring examples that might be too difficult to learn, perhaps because they are unrepresentative samples or they have have label noise.

## 5.2 Connections to the Linear Mode Connectivity

We now examine how the ranking of the examples by EL2N connects to the geometry of the loss surface. In particular, Frankle et al. [20] studied the effect of minibatch randomness on the training trajectory, focusing on identifying the point in training when two networks, starting from the same weights, but trained with independent minibatches, converge to the same "linearly connected" mode. They find that, for standard vision datasets, the onset of this "linear mode connectivity" (LMC) happens early in training.

More precisely, let $w_1, w_2, \ldots, w_T$ be the training trajectory of a *parent* network, fix a *spawning time $t^*$*, and let $v_{t^*}, v_{t^*+1}, v_{t^*+2}, \ldots, v_T$ be an independent training trajectory (i.e., with independent minibatches), beginning at $v_{t^*} = w_{t^*}$. We call $v_T$ the child network and $v_{t^*}, v_{t^*+1}, \ldots$ the child trajectory. The (training) error barrier between two weights $w$ and $w'$, denoted $\text{err}(w, w'; S)$, is the maximum deviation of the training error surface $\hat{R}_S(\cdot)$ above the line connecting the empirical risk at $w$ and $w'$. That is,

$$\text{err}(w, w'; S) = \sup_{\alpha \in [0,1]} \left\{ \hat{R}_S(\alpha\, w + (1 - \alpha)\, w') - \alpha\, \hat{R}_S(w) - (1 - \alpha)\, \hat{R}_S(w') \right\}. \quad (5)$$

We then define the *mean (training) error barrier, spawning at $t^*$, at time $t$*, for $t^* \leq t \leq T$, denoted $\text{err}_t^{t^*}(S)$, to be the expected error barrier between $w_t$ and $v_t$ on the data $S$. That is,

$$\text{err}_t^{t^*}(S) = \mathbb{E}_{w_{t^*+1:t}, v_{t^*+1,t}}[\text{err}(w_t, v_t; S)], \quad (6)$$

where the expectation is taken over the randomness in the trajectories of $w$ and $v$ *after $t^*$* due to the choice of minibatches, conditional on the initial trajectories up through time $t^*$. (Note that, at the end of training $t = T$, the supremum in $\text{err}(w_T, v_T; S)$ is often achieved near $\alpha = 1/2$, and so this is a cheap approximation used in practice.) The "onset" of linear mode connectivity is the earliest spawning time $t^*$ at which point $\text{err}_T^{t^*}(S) \approx 0$, where $S$ is the whole training set. In our work, we instead compute the error barrier on *subsets of the training set*, which allows us to compare the training dynamics and modes on *subpopulations*.

In Fig. 4, we measure the mean error barrier $\text{err}_t^{t^*}(S')$ as a function of the spawning time $t^*$, in the cases where $S'$ are either 1) the training examples with the smallest scores, 2) the largest scores, or 3) a random subset of training examples. We find that the error barrier falls close to zero very rapidly for examples that have low EL2N scores, and stays high for high score examples. These findings suggest that the loss landscape derived from restricted subsets of examples with low and high EL2N behave very differently. The loss landscape derived from easy subsets of examples with low scores is quite flat, in the sense that error barriers between children as a function of spawn time rapidly diminish. On the other hand, the loss landscape derived from harder subsets of examples with higher scores is rougher, with higher error barriers that persist for longer in the spawn time. Further, this result is in agreement with the results presented in Section 5.1, showing that most of the learning happens in the high EL2N score examples.

## 6 Related Work

As discussed earlier, our work is closely related to an empirical study by Toneva et al. [8], which examines the frequency with which correct classification decisions are forgotten during training.

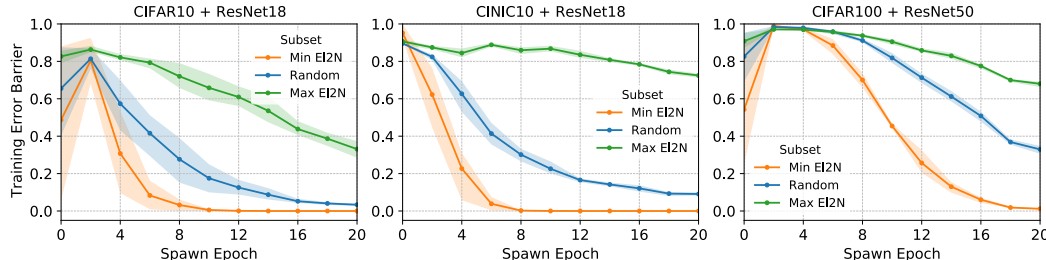

Figure 4: The final training error barrier between children on subsets of a 1000 highest (*green*) and lowest (*orange*) EL2N score examples, and randomly selected training subset (*blue*) as a function of the spawning time. *Left to right*: different dataset and network combinations.

The authors observe that examples that are rarely forgotten are also ones that do not contribute much to the final accuracy of the predictor. In particular, if we retrain from initialization after having removed rarely forgotten examples from the training data, we achieve the same accuracy. Similar to our work, this work analyzes the dynamics of training in deep learning through the lens of training examples. However, unlike forgetting scores, our proposed methods use only local information *early in training*. This highlights two properties: example importance is reflected in the local properties of the loss landscape after a few epochs and the ordering of examples by importance is roughly preserved throughout training. We think that this is a key contribution and hope it prompts future empirical and theoretical work exploring the early stage of learning.

Coleman et al. [1] use a small proxy network in combination with other training data selection methods to find a small subset of important-for-training examples, that can then be used to train a large state-of-the-art (SOTA) deep neural network. In their empirical study, they observe that most important examples selected via a proxy model are also important for training a SOTA network. In addition, they study a proxy which reuses SOTA network's architecture but is trained for a shorter time. The authors observe that selecting the important examples after at least 50 epochs of training works better than selecting them at random, but not as well as after the full training run. They do not study shorter training times for proxies or relate it to the training dynamics in any other way.

Another line of related work is on coresets (see, e.g., [4, 5, 7, 21–23], and many others). The term *coresets* generally refers to a possibly weighted subset of training data. Much of the work on coresets is focused on identifying small training data subsets that provably yield an $\epsilon$-approximate solution to the original objective (on all the training data). Most guarantees require the problem to have special structure, such as convexity. For nonconvex problems, like training deep neural networks, guarantees are provided for very conservative proxies, e.g., based on Lipschitz constants or smoothness. While coreset selection comes with nice theoretical guarantees, in our opinion, the utility of these methods is best considered an empirical question.

Coresets have also been studied in the active learning community. Here, the goal is to select a small set of examples to label at any given iteration of training (see, e.g., [24–28], and references therein). Coreset selection has also been proposed as a way to increase model robustness [29].

Pleiss et al. use a similar method, the Area Under the Margin (AUM) statistic, but with a slightly different goal: identifying noisy and mislabeled examples. Their proposed method exploits differences in the training dynamics of clean and mislabeled samples by keeping track of statistics through the course of training. The AUM statistic is similar to forgetting scores in that it uses information from the whole training run. In contrast, we focus on *instantaneous information* in the early phase of training. In addition to identifying noisy examples, we aim to rank points by importance and therefore also identify redundant/easy examples. Thus our approach is complimentary to the AUM statistic and can be used together to obtain higher levels of pruning on noisy datasets.

There have been a number of recent studies looking at the problem of estimating example difficulty. One such approach for identifying difficult examples within a given class is the Variance of Gradients (VoG) score proposed by Agarwal, D'souza, and Hooker [31]. For each image, they calculate the gradient of the activations with respect to the pixels at $K$ different checkpoints over training. The VoG score is the average (over pixels) of the per-pixel variance across these $K$ checkpoints. The authors conclude that the images that appear more difficult also have a higher VoG score. Un-

derstanding the connections between these two scores calculated in entirely different spaces is an interesting direction for future work.

Another quite different approach estimates example difficulty using prediction depth [32], which is defined as the first layer at which a k-Nearest Neighbor classifier can correctly classify an example using the representation of the image in all subsequent layers. In additional to methodological differences, this method uses the final trained network. To our knowledge, we are the first to highlight the existence of a strong signal for estimating example difficulty early in training.

Informally, removing a training example from the training data and not hurting the generalization error suggests that the example has small "influence" on the test data. Influence of the training examples on test examples is studied in sample-based explainability [33–35]. On the theory side, Feldman [15] recently proposed to model data as a mixture of populations and study the role of memorization when the data distribution is long-tailed. Feldman demonstrates conditions under which memorization is necessary for good generalization. In doing so, he proposes a definition of example memorization and influence, which can be interpreted as a leave-one-out notion of stability. In an empirical study following this work, Feldman and Zhang [36] demonstrate that classifiers trained on computer vision benchmarks benefit from memorization. In particular, training without high-memorization-value examples comes at a cost of accuracy of the learned neural network classifier. In Appendix G, we compare GraNd, EL2N, forgetting scores, and memorization values on CIFAR-100-trained Resnet50 networks; memorization values do not correlate with the other scores.

## 7 Discussion

In summary, our work both (1) introduces methods to significantly prune data without sacrificing test accuracy using *only* local information *very early* in training (Fig. 1), sometimes even at initialization, and (2) uses the resulting methods to obtain new scientific insights into how different subsets of training examples drive the dynamics of deep learning. We start from a principled approach by asking how much on average each training example influences the loss reduction of other examples, and from that starting point, we obtain 2 scores, namely gradient norm (GraNd) and error norm (EL2N) that bound or approximate this influence, with higher scores indicating higher potential influence. We find that examples with higher scores tend to be harder to learn, in the sense that they are forgotten more often over the entire course of training. We also find that the very highest scoring examples tend to be either unrepresentative outliers of a class, have non standard backgrounds or odd angles, are subject to label noise, or are otherwise difficult. This observation yields a simple and powerful sliding window method (Fig. 2) to prune data by keeping examples within a range of scores, where the start and the end of the range constitute just 2 hyperparmeters that can be tuned via a validation set. This tuning can be done using different hyperparameter settings or on a different network saving computation time (Appendix E.3). Furthermore, we find that high-scoring examples primarily drive feature learning by maximally supporting the velocity of the NTK, whereas learning dynamics might actually give up on the very highest scoring examples that may correspond to unrepresentative examples or noise (Fig. 3). Finally we show that higher (lower) scoring subsets of examples contribute to a rougher (smoother) loss landscape (Fig. 4). Overall this decomposition of both loss landscape geometry and learning dynamics into differential contributions from different types of examples constitutes an exciting new methodology for analyzing deep learning. A deeper understanding of the differential role played by different subsets of examples could aid not only in data pruning, but also in curriculum design, active learning, federated learning with privacy, and analysis of fairness and bias. Our empirical findings raise a number of interesting theoretical questions, some of which we discuss in Appendix D.

## Acknowledgements

The authors would like to thank Blair Bilodeau, Alex Drouin, Étienne Marcotte, and Daniel M. Roy for feedback on drafts, the Toolkit team at ServiceNow for providing the tools and computation resources that greatly accelerated our empirical work, and the NeurIPS reviewers for their thorough engagement and invaluable feedback on label-dependence and practical applications. S.G. thanks the Simons Foundation, NTT Research and an NSF Career award for funding while at Stanford.

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
