

Figure 5: Examples with the smallest (first row) and second smallest (second row) GraNd scores for each class (columns, from left to right: airplane, automobile, bird, cat, deer, dog, frog, horse, ship, truck) from a ResNet18 trained on CIFAR-10. GraNd scores were calculated at initialization.

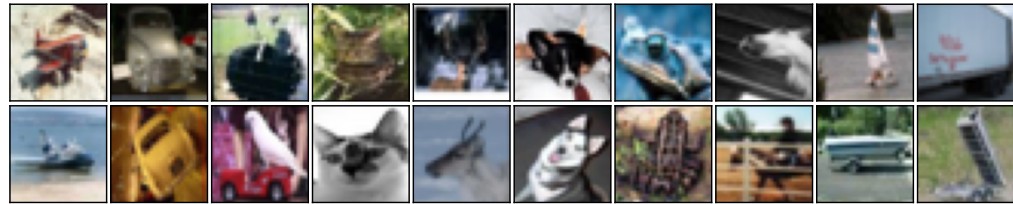

Figure 6: Examples with the second largest (first row) and largest (second row) GraNd scores for each class (columns, from left to right: airplane, automobile, bird, cat, deer, dog, frog, horse, ship, truck) from a ResNet18 trained on CIFAR-10. GraNd scores were calculated at initialization.

## A    Ethical and societal consequences

This work raises several ethical considerations. Being, an empirically driven work, it consumed considerable energy. However, we hope that it will enable advancements in theory that will more efficiently guide experiments. Also, we focus mostly on accuracy as a metric, which tends to hide disparate effects on marginalized groups. But since this work attempts to explicitly uncover the influence of training examples and sub-populations, we hope that it will lead to methods that will decrease bias in the training procedure, especially if marginalized groups are under-represented in the dataset and are thus difficult to learn.

## B    Implementation Details

The code is made available at https://github.com/mansheej/data_diet

### B.1    Resources used

We run all experiments on a single 16GB NVIDIA Tesla V100 GPU. The entire project (from early exploration to final paper) used about 15000 GPU hours. We used an internal cluster at ServiceNow.

### B.2    Training details

**Deep Learning Frameworks.**    We used JAX [37] and Flax [38] in our implementations.

**Data.**    We use CIFAR-10, CIFAR-100 [12], and CINIC-10 [13]. CIFAR-10 and CIFAR-100 are used in their standard format. For CINIC-10, we combine the training and validation sets into a single training set with 180000 images. The standard test set of 90000 images is used for testing. Each dataset is normalized by its per channel mean and standard deviation over the training set. All datasets get the same data augmentation: pad by 4 pixels on all sides, random crop to $32\times32$ pixels, and left-right flip image with probability half.

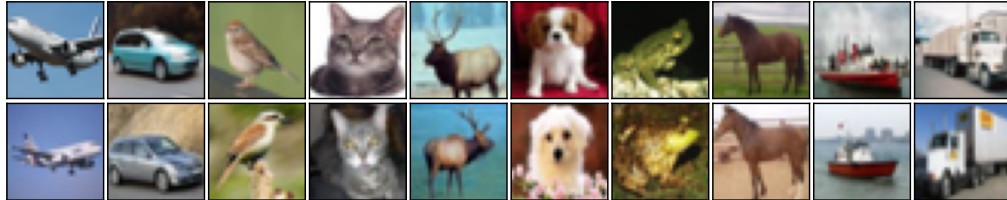

Figure 7: Examples with the smallest (first row) and second smallest (second row) EL2N scores for each class (columns, from left to right: airplane, automobile, bird, cat, deer, dog, frog, horse, ship, truck) from a ResNet18 trained on CIFAR-10. EL2N scores were calculated at epoch 10.

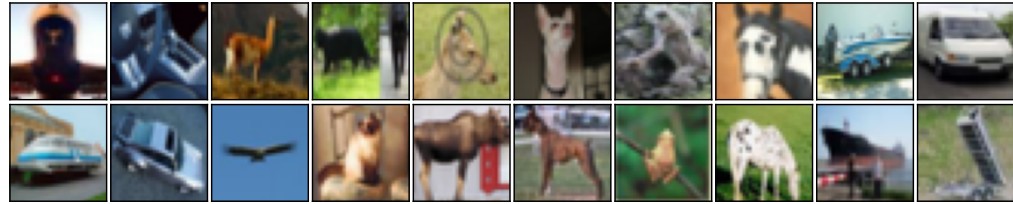

Figure 8: Examples with the second largest (first row) and largest (second row) EL2N scores for each class (columns, from left to right: airplane, automobile, bird, cat, deer, dog, frog, horse, ship, truck) from a ResNet18 trained on CIFAR-10. EL2N scores were calculated at epoch 10.

**Models.** We use ResNet18-v1 and ResNet50-v1. Our implementation is based on the example in Flax [38] designed for larger high resolution images. Since CIFAR and CINIC images are 32×32 pixels only, we use the low-resolution variant of these networks: the first two layers (a convolution layer with 7×7 kernel and 2×2 stride, and a max pooling layer with 3×3 kernel and 2×2 stride) are replaced with a single convolution layer with 3×3 kernel and 1×1 stride.

**Training hyperparameters.** All networks are trained with the Stochastic Gradient Descent (SGD) optimizer, learning rate = 0.1, nesterov momentum = 0.9, weight decay = 0.0005. For CIFAR-10 and CIFAR-100, we use batch size = 128, and for CINIC-10, we use batch size = 256. The learning rate is decayed by a factor of 5 after 60, 120 and 160 epochs and all networks are trained for a total of 200 epochs (for the full dataset, i.e. 78000 steps for CIFAR-10 and CIFAR-100, and 140600 steps for CINIC-10). When using a pruned dataset, to allow for a fair comparison with the full dataset, we keep the number of iterations and schedule fixed for different pruning levels.

### B.3 Experimental details

**Reporting results.** For every quantity we plot, we do 4 independent runs (independent model initialization and SGD noise) and report the mean and the 16th to 84th percentile of obtained accuracies for representing variability across runs. The mean is reported as lines and the variability is reported as shading in the plots. When evaluating scores, we train new randomly initialized models with different seeds from those used to calculate the scores.

**Calculating scores.** All scores (EL2N, GraNd and forgetting scores) are calculated by averaging the scores across 10 independent runs.

**Random label experiments.** For the random label experiments, at the beginning of training, we pick 10% of the examples randomly and permute their labels (to keep overall label statistics fixed). The subsets are selected as follows:

1. score all examples and sort them in ascending order by score;
2. drop the set of images with the smallest scores that make up a fraction of the dataset equal to the specified offset;

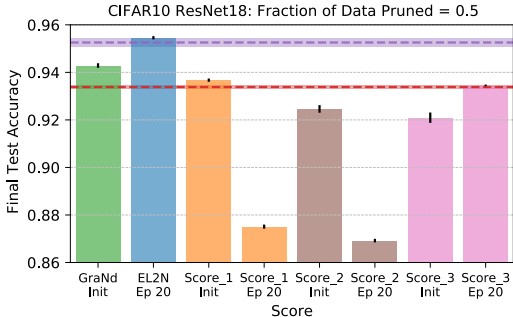

Figure 9: Compare GraNd and EL2N scores (which are label dependent) to three label independent variants at initialization and early in training. Experiments performed on ResNet18 trained on CIFAR10. Purple dashed line corresponds to final test accuracy after training on the full training dataset and red dashed line corresponds to final test accuracy after training on a random 50% subset of the data. Bars are the final test accuracies obtained from training on 50% of the training data with the highest corresponding score. Definitions of scores 1-3 in Appendix D.

3. keep the next set of images based on the given subset size;

4. drop all the following images.

**Kernel velocity experiments.** The kernel velocities are calculated with 100 examples. The examples are picked as follows: first, score all examples and sort them in ascending order by score; then, for every example index in the figure, calculate the NTK gram matrix velocity (described in Section 5.1) for 100 contiguous sorted images starting at that example index.

The NTK submatrix velocity on a subset of examples at a particular point in training is defined as follows. Let the examples be $x_1, x_2, ..., x_m$. Let the time at which the NTK submatrix is calculated be $t$, the parameters of the network $f$ is $\mathbf{w}_t$. Let $C$ be the number of classes and $N$ the number of parameters in the network. Using the notation in Section 2.1, the $k$th logit gradient of the model on example $i$ is $\psi_t^{(k)}(x_i) = \nabla_{\mathbf{w}_t} f_t^{(k)}(x_i)$. We cast these into a $mC \times N$ matrix $\Psi_t$ where the rows run over each logit gradient of each image and the columns run over the parameters of the model. The NTK submatrix is a $mC \times mC$ matrix given by $K_t = \Psi_t \Psi_t^T$. The kernel velocity is calculated as

$$v = 1 - \frac{\langle K_t, K_{t+1}\rangle}{\|K_t\|\|K_{t+1}\|} \tag{7}$$

where $\langle \cdot, \cdot \rangle$ is the Frobenius inner product and $\|\cdot\|$ is the Frobenius norm.

**Linear mode connectivity experiments.** The training error barrier between children is calculated by following [9, 20]. In addition to estimating the error barrier on 1000 random images, we also estimate the error barrier on 1000 images with the largest and smallest EL2N score. The EL2N score used is calculated at epoch 10 of the parent run.

## C   Example Images

In this section, we examine the examples with small and large GraNd and EL2N scores for a ResNet18 trained on CIFAR-10. GraNd scores were computed at initialization and EL2N scores at epoch 10. We show two examples from each class with both minimum and maximum EL2N and GraNd scores in Figs. 5 to 8. The examples with the minimum GraNd and EL2N scores tend to be simple, canonical representations of each class pictured from very typical angles. The examples with maximum scores are harder to identify; they are blurrier, from strange angles or have unexpected backgrounds or other artifacts.

## D    Comparisons to Label Independent Scores

For classification tasks, the GraNd score for a training example $(x, y)$, is defined as

$$\text{GraNd}_t(x, y) = \mathbb{E} \left\| \sum_{k=1}^{K} \left( p(\mathbf{w}_t, x)^{(k)} - y_k \right) \psi_t^{(k)}(x) \right\|_2 \tag{8}$$

where $k$ indexes classes.

At initialization, the weights are label independent, and thus $\psi_t^{(k)}(x)$ has no label information. Therefore, the GraNd score in theory should not depend on the labels.

Note, however, that in practice we approximate the expectation using a finite number of samples of different initializations. In this section we ablate the GraNd score to tease apart the contribution of the labels, if any, and whether there is any signal in the logit gradient averaged over classes.

In contrast to the GraNd score, the EL2N scores consist of just the label prediction error:

$$\text{EL2N}_t(x, y) = \mathbb{E} \left\| p(\mathbf{w}_t, x) - y \right\|_2 . \tag{9}$$

We evaluate three different label independent variants, defined as

$$\text{Score}_{1t}(x) = \mathbb{E} \left\| \sum_{k=1}^{K} \psi_t^{(k)}(x) \right\|_2 , \tag{10}$$

$$\text{Score}_{2t}(x) = \mathbb{E} \left\| p(\mathbf{w}_t, x) - \frac{1}{K} \mathbf{1} \right\|_2 , \tag{11}$$

$$\text{Score}_{3t}(x) = \mathbb{E} \left[ \sum_{k=1}^{K} \| \psi_t^{(k)}(x) \|_2^2 \right] . \tag{12}$$

Our experimental setup is as follows:

1. Train 10 independent ResNet18 networks on CIFAR10.
2. Calculate the scores in Eqs. (10) to (12) at both initialization and epoch 20. The expectation is taken across the 10 networks. This gives us 6 new scores: 3 score types at 2 initializations each.
3. Prune 50% of the CIFAR10 training examples that have the smallest scores. This gives us 6 new pruned datasets, one for each score.
4. Train 3 new randomly initialized ResNet18 networks on each of the 6 pruned datasets.
5. For each of the 6 scores, evaluate their performance by averaging their final test accuracy across the 3 networks trained on the corresponding dataset.

In Fig. 9 we compare these scores to training with a 50% subset of maximum GraNd scores at initialization, maximum EL2N scores at epoch 20, a random 50% subset and the full training set. Only the label-dependent scores perform significantly better than the random baseline.

One unanticipated finding was that the GraNd score outperformed $\text{Score}_1$, it's label-independent variant, at initialization. This empirical observation presents a discrepancy between the score as defined in Eq. (9), and it's empirical version. This discrepancy suggests some interesting directions for future research. A mathematical theory of how the GraNd score behaves with respect to labels when averaging over a small number of initializations could potentially inspire new improved methods.

## E    Additional Experiments

### E.1    Sensitivity analysis of GraNd and EL2N scores

In all our experiments, GraNd and EL2N scores are averaged over 10 independent initializations or runs. This turns out to be essential for successful pruning using the scores.

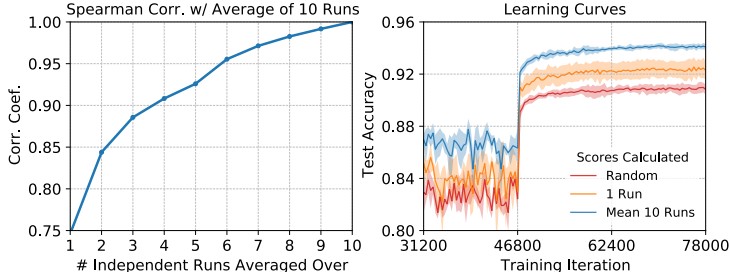

Figure 10: The effect of averaging GraNd scores calculated at initialization of a ResNet18 trained on CIFAR-10. *Left:* The Spearman rank correlation coefficient between GraNd scores obtained by averaging over a given number of independent runs (x-axis) to those obtained by averaging over 10 independent runs. *Right:* Training accuracy curves for ResNet18 trained on a 50% subset of CIFAR-10. The training subset is obtained by either random sampling or keeping the examples with the largest GraNd scores. We compare the test accuracy obtained using GraNd scores from 1 initialization and from averaging over 10 independent initializations. We zoom in to the end of training to highlight the differences between the learning curves.

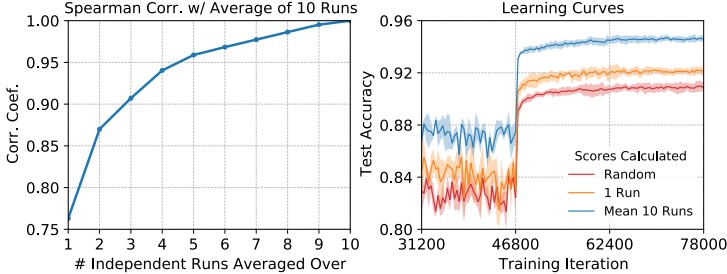

Figure 11: The effect of averaging EL2N scores calculated at epoch 10 of a ResNet18 trained on CIFAR-10. *Left:* The Spearman rank correlation coefficient between EL2N scores obtained by averaging over a given number of independent runs (x-axis) to those obtained by averaging over 10 independent runs. *Right:* Training accuracy curves for ResNet18 trained on a 50% subset of CIFAR-10. The subset is selected by either random sampling or keeping the examples with the largest EL2N scores. We compare the test accuracy obtained when pruning based on EL2N scores from a single run and from averaging over 10 independent runs. We zoom in to the end of training to highlight the differences between the learning curves.

In Fig. 10 (right) we show the effect of averaging on GraNd scores computed at initialization on a ResNet18, CIFAR-10. On average, GraNd scores for any individual run have a Spearman rank correlation of about 0.75 with the GraNd scores averaged over 10 runs.

We also compare learning curves when training using 50% of the training data selected as follows: using high GraNd scores at a single initialization; using high GraNd scores averaged over 10 initializations; randomly (baseline). As seen in Fig. 10 (right), using scores averaged over 10 initializations performs significantly better. In Fig. 11, we show similar results for EL2N scores calculated at epoch 10 on a ResNet18 trained on CIFAR-10.

These results suggest that GraNd and EL2N scores represent properties of the dataset rather than of a specific network weights. To get an accurate ranking of example importance, we need to average out the effects of individual initializations/weights. Empirically, we find that averaging over 10-20 runs suffices, and averaging over more runs has insignificant additional benefit.

### E.2 Comparison between scores from different architectures on the same dataset

In this section, we examine how the choice of network architecture affects pruning with EL2N and GraNd scores. Specifically, we repeat the experiment in Fig. 1 bottom row, but for a ResNet18 and a ResNet50 trained on CIFAR-10. The results are shown in Fig. 12. Both networks share the

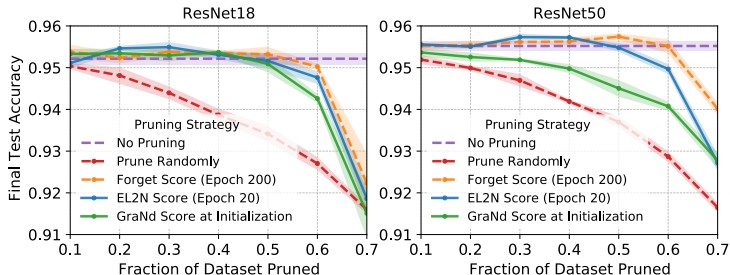

Figure 12: Experimental setup similar to Fig. 1. We compare CIFAR10 data pruning on ResNet18 and ResNet50. The y-axis indicates the final test accuracy achieved by training after pruning different fractions of the dataset (x-axis). Compare forgetting scores at the end of training, EL2N scores early in training (at epoch 20) and GraNd scores at initialization. In each case, examples with the lowest scores are pruned and then the networks are trained from initialization on the data that was not pruned. In both plots accuracy achieved by training on the full dataset and on a random subset of the corresponding size dataset are used as baselines.

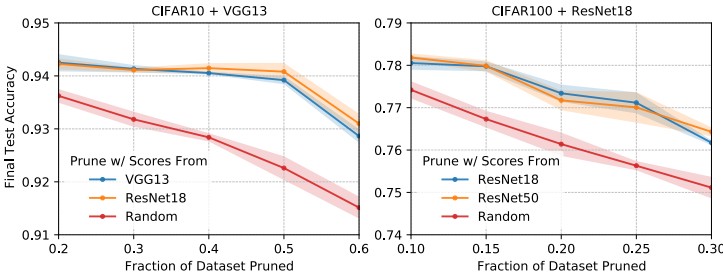

Figure 13: EL2N scores calculated using one network can be used to prune data for training with another network. Left: we compare VGG13 networks trained on CIFAR10 after pruning different fractions of the train set. The examples with the lowest EL2N scores were dropped. EL2N scores were calculated at epoch 20 using 10 independent networks, with either a VGG13 or a ResNet18 architecture. Right: we compare ResNet18 networks trained on CIFAR100 after pruning different fractions of the train set. The examples with the lowest EL2N scores were dropped. EL2N scores were calculated at epoch 20 using 10 independent networks, with either a ResNet18 or a ResNet50 architecture. Random pruning is used as a baseline. Test accuracy after pruning is agnostic to which network was used to calculate the scores: EL2N scores generalize across architectures.

same overall patterns: pruning by GraNd or EL2N scores before training results in roughly the same accuracy for different levels of pruning. Further, independently of the network architecture tested, data pruning by EL2N scores computed at epoch 20 is competitive with pruning based on forgetting scores computed at epoch 200. When trained on either of the networks, pruning based on GraNd scores does significantly better than the random baseline. Overall, these results suggest that network depth has a small effect on our data-pruning results.

Surprisingly, GraNd scores at initialization for ResNet18 seem to be better for pruning than those for ResNet50. In future work, we hope to better understand the relationship between the level of pruning and network depth and width.

### E.3  EL2N scores generalize across architectures

EL2N scores are robust to variations in architectures and seem to capture information intrinsic to the dataset, not network specific behavior. To support this, we show that EL2N scores calculated with one architecture can be used to prune the dataset for other architectures; scores calculated from a different architecture leads to final test accuracies that are identical to those trained using scores calculated from the same architecture. Our experiment setup is as follows:

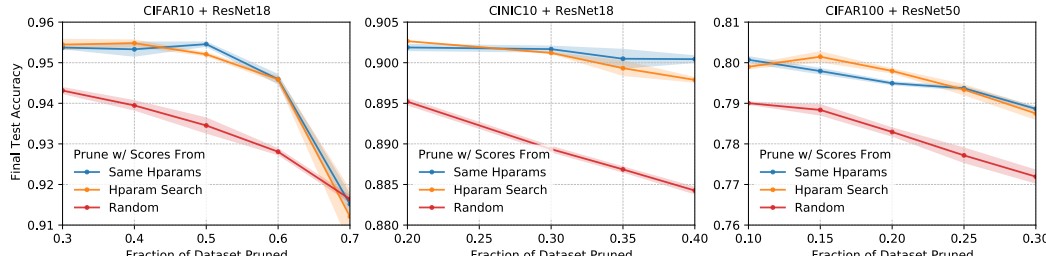

Figure 14: EL2N scores can be effectively calculated using networks from hyperparameter search runs. In this experiment, we calculate EL2N scores by averaging over networks trained with different hyperparameter configurations. Hyperparameters were chosen for a grid search. We compare this to EL2N scores calculated by averaging over networks trained with the optimal hyperparameters. In both cases, scores were calculated at epoch 20. The scores are evaluated by comparing the final accuracy of networks trained on datasets pruned by either score to different sizes. The evaluation networks were trained with optimal hyperparameters. We also compare to a random pruning baseline. From left to right, we perform this experiment for CIFAR10 with ResNet18, CINIC10 with ResNet18 and CIFAR100 with ResNet50. In all cases, for a wide range of pruning percentages, EL2N scores calculated during hyperparameter optimization perform as well as scores calculated using optimal hyperparameters.

1. For CIFAR10, calculate the EL2N scores at epoch 20 using a ResNet18 and a VGG13. This is done by averaging over 10 independently initialized networks.

2. Use both scores to train new randomly initialized VGG13 networks on n% subsets of CIFAR10, keeping only the highest scores for each score type. We perform this experiment for a range of n (n = 20, 30, 40, 50, and 60).

VGG13 networks trained on subsets using either score perform identically and significantly better than random. To show robustness across architectures and datasets, we repeat this experiment for CIFAR100, ResNet18 instead of VGG13 and ResNet50 instead of ResNet18 with n = 10, 15, 20, 25, 30. Our results in Fig. 13 show that, for a given network and datasets, pruning performance of EL2N scores are agnostic to which network is used to calculate them. In addition to being a tool that extracts something intrinsic about the dataset, this has practical applications as, for a given dataset, one could extract a smaller important subset for future networks to train on, thereby speeding up the iteration process.

### E.4 Calculating EL2N scores during hyperparameter optimization

For already benchmarked datasets, it can be helpful to reduce the size of the dataset for training future networks. But in practice, we often have to train on new datasets, and for each new dataset, we need to perform a hyperparameter optimization search. In the next experiment, we show that networks trained during hyperparameter optimization can be used to calculate EL2N scores that perform as well as EL2N scores calculated by averaging over an ensemble of networks with optimal hyperparameters. This is a robust effect across architectures and datasets, we show it for ResNet18 traind on CIFAR10, ResNet18 trained on CINIC10, and ResNet50 trained on CIFAR100. Our experiment setup is as follows:

1. For each architecture and dataset configuration perform a hyperparameter grid search over learning rates in $\{0.2, 0.1, 0.05\}$ and weight decays in $\{0.001, 0.0005, 0.0001\}$. This leads to 9 training runs using different hyperparameters for each configuration.

2. For each dataset, calculate EL2N scores at epoch 20 by averaging over the 9 corresponding networks.

3. For each dataset, prune a range of fractions of the data by dropping examples with the lowest corresponding score.

4. For each architecture and dataset configuration, train new randomly initialized networks with optimal hyperparameters on the pruned datasets and evaluate their performance.

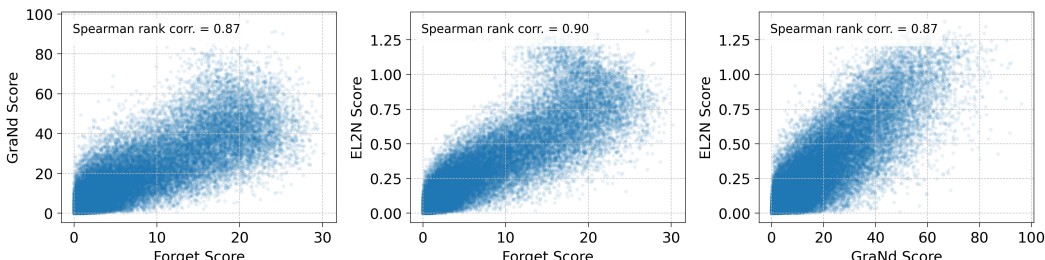

Figure 15: For a ResNet18 trained on CIFAR-10, we compare the values and Spearman rank correlations for pairs of scores. The scores compared are GraNd scores at initialization, EL2N scores at epoch 20 and forgetting scores at epoch 200.

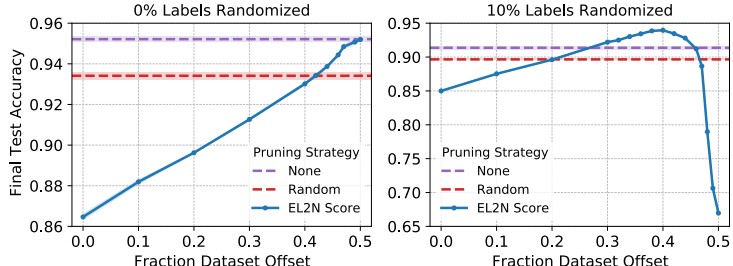

Figure 16: ResNet18 trained on a 50% subset of CIFAR-10 with clean *(left)* and 10% randomized labels *(right)*. The training subset contains the *lowest* scoring examples *after* examples with scores below the offset (x-axis) are discarded. Scores computed at epoch 10.

As baselines, we compare to random subset baselines and to EL2N scores at Epoch 20 calculated by averaging over 9 networks trained with optimal hyperparameters. Our results are in Fig. 14. For all three architecture and dataset configurations, and for a range of data pruning levels, EL2N scores calculated during hyperparameter optimization perform as well as EL2N scores calculated using networks with optimal hyperparameters. This further demonstrates that EL2N scores capture intrinsic dataset properties and not network specific properties. It also has the nice benefit that, after hyperparameter optimization, we essentially get EL2N scores for the price of one forward pass through the whole dataset, thus reusing the expensive compute from hyperparameter search to reduce future compute and make network training more energy efficient. An interesting direction for future research is to cleverly use scores to identify smaller datasets that can be used to speed up the hyperparameter optimization process.

### E.5 Correlations between scores

As discussed in previous sections, we find that different scores lead to similar pruning results. In Fig. 15, for a ResNet18 trained on CIFAR-10, we compare the values and Spearman rank correlations for pairs of scores. GraNd scores are computed at initialization, EL2N scores at epoch 20 and forgetting scores at epoch 200. The scores have high Spearman rank correlation with each other. EL2N and forgetting scores, which have the most similar performance, have the highest Spearman rank correlation.

## F Noise

### F.1 Noisy Examples in Low Pruning Regime

We repeat the noisy labels experiment of Fig. 2 in a different regime; instead of pruning 60% of the data, we prune just 50% of the data. Results are shown in Fig. 16. For the experiment with no labels randomized, at this lower pruning level we no longer see a boost in performance from dropping the highest score examples. However we do see decreasing marginal gains. This suggests that when we

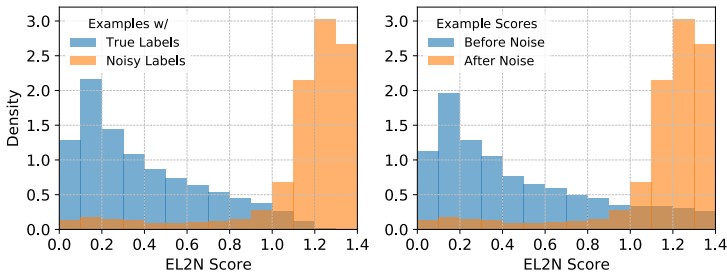

Figure 17: Scores for examples in the noisy label experiment of Fig. 2 *Left.* Distribution of EL2N scores for examples with true labels and corrupted labels. *Right.* Distribution of EL2N scores for examples that are corrupted before and after noise is added.

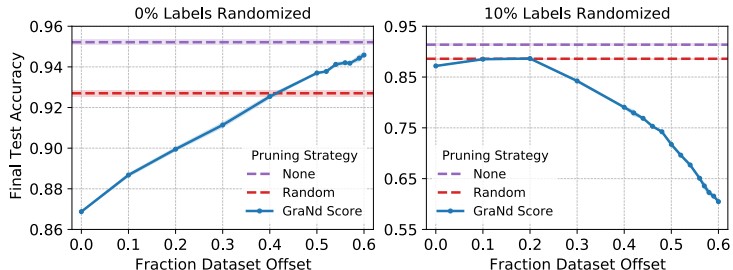

Figure 18: ResNet18 trained on a 40% subset of CIFAR-10 with clean *(left)* and 10% randomized labels *(right)*. The training subset contains the *lowest* scoring examples *after* examples with scores below the offset are discarded. Use GraNd scores computed at initialization.

have enough data, keeping the high score examples, which are often noisy or difficult, does not hurt performance and can only help.

## F.2    Scores for Noisy Examples

We now examine how adding noise affects the EL2N scores of examples. This analysis is done for the experiment in Fig. 2. See Section 4 for details. Results are shown in Fig. 17. Two results suggest that EL2N scores can be used to identify images with corrupted labels. First, the EL2N scores of images with corrupted labels tend to be higher than those with regular labels. Second, after an example's label is corrupted, its EL2N scores tends to be larger than before.

## F.3    Noise and GraNd Scores

We repeat the experiment from Fig. 2 (Section 4) except, instead of using EL2N scores at epoch 10, we use GraNd scores at initialization for pruning. As seen in Fig. 18, when none of the labels are corrupted, the results are similar to the previous variant of this experiment; keeping examples with larger GraNd scores in the training subset leads to better generalization performance. However, in the case where 10% of the labels are corrupted, we see the opposite trend: a model trained on larger GraNd scores ends up with lower accuracy, performing even worse than the random subset baseline. Note that the GraNd scores are calculated at initialization with a randomly initialized network. These results suggest that GraNd scores at initialization successfully find important examples only because of some favorable properties of the data distribution; adding noise to samples selected uniformly over the training set cripples the method. In future work, we further explore how high Bayes error rate effects GraNd scores at initialization.

## G    Comparison to memorization threshold

In Fig. 19 we compare EL2N scores to the memorization values defined in [36]. The memorization values for 1015 CIFAR100 examples are provided by the authors. We replicate their setting by

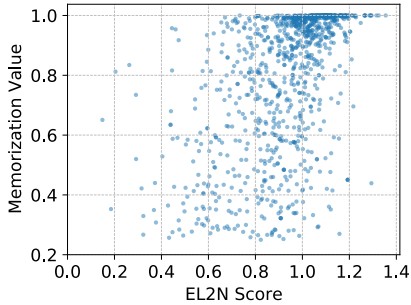

Figure 19: Comparison of EL2N scores and Memorization values from [36]

training a ResNet50 on CIFAR-100 and compute the EL2N scores at epoch 20 for the 1015 examples they provide. As seen from Fig. 19, memorization values and EL2N scores do not appear to be correlated.