# OpenReview forum: "Deep Learning on a Data Diet: Finding Important Examples Early in Training"
_NeurIPS.cc/2021/Conference — NeurIPS 2021 Poster_

### Official Review · Reviewer_V5c3 · 2021-07-12

**Rating:** 6
**Confidence:** 4

**Summary:**

The paper focuses on identifying important examples for training neutral networks early in training with the goal of pruning a significant portion of training examples in later training stages. It analyses the influence of a training example on the loss reduction of other examples and defines two scores:  gradient norm (GraNd) and error norm (EL2N) that bound or approximate this influence. It shows that the  higher scores generally correlate with the higher influence. However, the paper also finds that the highest scoring examples tend to be outliers and are subject to label noise, or are otherwise difficult. The paper then proceeds with analyzing the high and low scoring examples (under the defined GraNd and EL2N metrics) in training dynamics.


**Limitations And Societal Impact:**

yes

**Main Review:**

On one hand, the findings of this paper seem quite trivial, as examples that have larger gradients (in expectation) are likely to be more important. It reminds the heuristics of choosing neurons with higher weigh norm in neural network pruning in earlier works on model compression. On the other hand, the intuitive rule in this paper is motivated by a principled analysis (or at least an attempt to derive one). My other concern is the practical value of this paper. The empirical results show that when the pruned ratio becomes interesting (over 50%) then the proposed data pruning method is only marginally better than a random pruning.

**Time Spent Reviewing:**

2

---

> ### Author Response · Authors · 2021-08-10
> **Response to reviewer V5c3**
>
> Thank you for your review! We appreciate your concerns and make a case for why we think that this method, though seemingly trivial, is actually quite non-trivial. We also elaborate on potential practical applications and the interesting and important questions that this work poses for future experimental and theoretical research. We look forward to further discussions about our work with you.
>
> The review suggests that our scores, at least on the surface, seem quite trivial—examples with larger gradients, on average, are likely to be important. However, this is not obvious; there is no reason to a priori expect that the gradient norms at initialization, or even the error vector a handful of epochs into training would be useful for studying the outcome of *thousands of iterations of non convex optimization*. To understand this, consider the context of this work.
>
> Fundamentally, we are interested in finding and keeping only important training examples—examples that a network must see in order to achieve optimal test error at the end of training. More precisely, consider training a network *to completion* using *all* the training data. Then remove a single training example, and retrain the network from scratch using all the data *except this* example, while averaging over initializations and SGD noise. If the test loss without this example increases a lot, relative to training with all the data, then this example is by definition more important. On the other hand, if the test loss does not increase very much, it is by definition unimportant (one can drop it from the training set without sacrificing much test accuracy). One can then sort all the examples by their importance. Of course, while this is a rigorous and intuitive definition of example importance, it is *completely computationally intractable*.
>
> Framed in this context, the question is more challenging—how does one find examples that are important for achieving high test accuracy after thousands of iterations of non convex optimization, without the benefit of observing the trajectory? How can one do this in a computationally tractable manner using information only near the beginning of training or even only at initialization? Before our work, it was not at all obvious how to do this. Other groups whom we have spoken to have tried and failed and were extremely surprised by our ability to succeed in this endeavor. Before our experiments we doubt anyone would have seriously suggested that simply computing the gradient norm of each example at initialization and averaging over initializations is enough to approximate example importance. We thus disagree strongly with the reviewer that our findings are trivial.
>
> Moreover, the "obvious" intuition isn't correct and misses the important contribution that made this result possible—averaging over multiple initializations. As we show in Figs. 9-10, pruning based on scores calculated from one run, *even when the network we train on the pruned dataset has the same initialization as the one used to calculate the score*, performs significantly worse and doesn't obtain accuracies close to training on the full dataset. We averaged the scores across initializations in an attempt to "average out network specific details" and estimate the underlying example difficulty as a property of the dataset. This turned out to be the key step that seems to have been missed in the past leading to such a simple score being overlooked as a way to prune data in modern deep learning.
>
> It’s worth noting that one could also label forget scores as trivial: of course examples whose classification never changes during training are not influencing training. But it is actually an important observation that has spurred follow-up research, including this work.
>
> In addition to efficiently pruning data, we make new observations about how different examples with different importance scores influence training dynamics: higher scoring examples drive more change in the network’s representation, as measured by kernel velocity and higher (lower) scoring examples contribute more roughness (smoothness) to the loss landscape. These results are new and fundamental contributions to our scientific understanding of deep learning and connect our work to a large body of work examining deep learning through the lens of neural tangent kernels and loss landscapes. The simplicity of our method is a strength here: it brings a new perspective of data importance and can be used as a tool in future experimental and theoretical research on the connection between properties of the data distribution and training dynamics.
>
> We now switch gears to the practical value of this paper. Computationally efficient data pruning has practical applications even in the regime of pruning we are in. In addition to the obvious applications to reducing training time and memory, some non-obvious areas of impact are.
>
> - **Efficiency:** We ran experiments to verify that EL2N scores computed on one convolutional neural network (CNN) architecture, such as ResNet18, can be used to effectively prune data for other CNN architectures, such as VGG13 or ResNet50; these scores generalize across CNNs. Thus, as pointed out by reviewer EkuL, we can use these scores to produce an architecture-agnostic ranking of images for each dataset. Additionally, EL2N scores calculated by averaging over different training runs used for hyperparameter tuning are as effective for pruning as scores calculated by averaging over multiple runs with the same hyperparameters. Since, in most practical applications with new datasets, the optimal hyperparameters are not known and it is common to do multiple (partial) training runs for hyperparameter tuning, using our technique we get example importance at little extra cost. We will add these experiments to the appendix of our paper.
> - **Explainability/fairness/privacy:** Since our method requires little extra cost to what we already pay for hyper tuning, it has implications for data security: we can identify which points need to be kept and delete the rest. Also, our scores induce a ranking of training samples that could help identify examples that are rare or outliers. This information can be used when debugging or designing more reliable models. This is an avenue for future research.
> - **Inspiring new methods:** We are still at a stage where too little is known about what is possible in data pruning to be overly focused on achieving ideal results. A new benchmark showing that, early in training and with very simple metrics, we can achieve results similar to pruning at the end of training is valuable. It stands to reason that our work will inspire others to develop new methods to prune data early in training, potentially unleashing even greater gains. Additionally, we think that, used in creative ways, our method can speed up hyperparameter tuning, where the effects of compressing the dataset will add up.
> - **Inspire new theory:** Last but not least, we think that one of our key contributions is identifying and studying phenomena in deep learning training (“nice” properties of the training data, connections between training data and SGD trajectory, linear mode connectivity, NTK evolution). This work points the way to exciting new questions, including questions around the limits of data pruning early in training. We believe that our work will inspire and influence the development of theory that explains deep learning practice, in particular, the heterogeneous way in which data influences optimization/learning.

---

> > ### Comment · Reviewer_V5c3 · 2021-08-25
> > **Reply to rebattal**
> >
> > I thank the authors for the clarifications. The efficiency is still an issue, but I like the overall merit.

---

> > > ### Author Response · Authors · 2021-08-31
> > > **On efficiency**
> > >
> > > Sorry for the late response, but we only just noticed your comment here. We are glad to hear that our clarifications were useful. Regarding efficiency, we believe that we, as a community, are too early in the scientific process of understanding data pruning to be too focused on efficiency. It is useful to compare the situation to weight pruning: the situation there is that the algorithms are WAY less efficient. E.g., iterative magnitude pruning relies on the network being trained many times. Still, the hope is that studying these algorithms may yield insight that could lead to efficient techniques (or reveal fundamental barriers that prevent us). The same is true here. We are still at the stage of building our understanding. We happen to get a reasonably efficient algorithm out of it from the start, but this is just the beginning.
> > >
> > > If there are any other concerns that are preventing you from raising a score, please let us know and we will attempt to address them. However, if you think this work should be accepted, it will likely be necessary for you to raise your score to a straight "Accept".

---

### Official Review · Reviewer_EkuL · 2021-07-16

**Rating:** 6
**Confidence:** 3

**Summary:**

The paper proposes two metrics to rank training set samples for reducing the training set size:

1. L2-norm of the error vector (EL2N).
2. L2-norm of the gradient vector (GraNd).

These can be evaluated cheaply and relatively early in training (10-20 epochs) or even at initialization. For instance, by keeping only the top 50% CIFAR10 of training data according to this metric, one can achieve the same test accuracy as when using the whole training set.

The authors evaluate their method on several datasets and architectures, and run several experiments that relate these metrics to change in NTK during training or the onset of linear mode connectivity, and more.


**Limitations And Societal Impact:**

All good.

**Main Review:**

## Post-rebuttal update:


I am raising my score from 5 to 6 due to good results on architecture/hyperparameter robustness, and conditioned on other changes discussed in the rebuttal: ablation against label-independent metrics + impact of number of evaluations on performance of GraNd/EL2N + rework presentation / theory based on the mismatch between empirical and theoretical GraNd. I believe these changes will not change the paper results/narrative by too much, but will improve its significance and address some of my prior concerns about motivation/ablations. I thank the authors for actively engaging in the rebuttal.

-------
## Initial Review:

I find the key result of this paper very interesting and surprising (that one can trim the training set so significantly with such simple metrics). However, due to lacking motivation / insight / ablations, I do not feel like advocating for acceptance.

## Originality

To my knowledge the idea is novel, although I am not well familiar with dataset pruning literature.

## Quality

Overall the quality is good, and experiments are thorough and exhaustive. However, certain aspects are lacking that make me lower the score:

1. IMO section 2.2. is not at all clear and contains multiple errors. For instance:

	* Line 114: why does $\Delta_t$ depend on  $S_t$? I don't see it on the RHS. $f_t$ does not depend on $S_t$ per equation (1). But even if it did, extra dependence on $S_t$ would be redundant due to dependence on $f_t$.
    * Line 115: "where the gradient is computed on the minibatch $S_t$" - what does this mean exactly? Expression in line 114 is the derivative of the loss of outputs at $x$ after $t$ training steps, and as before, I don't see dependence on $S_t$.
    * Line 116: should it be $w_t - w_{t - 1}$? Otherwise RHS contradicts the definition in Equation (1).
    * Line 118: Here and after $\Delta_t$ is evaluated at the whole $S$, but I still don't understand how exactly it depends on this argument.
    * Line 120: Formula for $dw_t/dt$ now contradicts the one in line 116, where the sum is over $S_{t-1}$.
    * Line 124: "constant $c$ does not depend on $(x, y)$" - isn't it clearly defined as a function of $(x, y)$ in line 121?
    * Lines 131-133, 135-136: why do individual logit gradients become roughly orthogonal and similar in size _after_ training? I can see them being such at the beginning, but shouldn't logits start to differentiate as the network learns different classes?


Overall, I found this section quite vague and confusing, and hope it can be improved in terms of mathematical rigor and clarity.

2. By definition (i.e. in expectation over random initializations), for a fixed $x$, EL2N and GraNd at $t = 0$ are precisely the same for any $y$ among one-hot targets $[1, 0,\dots, 0],\dots, [0,\dots, 0, 1]$ via a symmetry argument. Therefore, at initialization, they can't possibly depend on $y$, and yet the metrics are still helpful per Figure 1 and others. Hence,

	2.1. I think it's necessary to ablate the metrics against $y$-independent baselines such as (a) the L2 norm of the network outputs $f_t(x)$ (or L2 distance from $f_t(x)$ to e.g. a constant vector of $[1/C,\dots,1/C]$), and (b) the Frobenius norm of the outputs-to-parameter Jacobian $\partial f_t(x)/\partial w_t$, or other alternatives. These could be simpler and more interpretable, and with more potential applications, since you don't need the label.

    2.2. Could you please confirm that the 10 random seeds used to compute the metrics do not intersect with 4 random seeds used to train and evaluate networks afterwards? Otherwise $y$ could indeed be useful at initialization, but only because it helps pick luckier random seeds.

## Clarity

Apart from section 2.2, the paper is very clearly written. However, in my opinion the experimental design itself doesn't shed that much light into the nature of these metrics, and they remain a bit mysterious to me. For instance, why is one sometimes better than the other? Why can there be dips in performance, as in e.g. epoch 2, Figure 1? Do training samples get reordered frequently as training progresses? Do these metrics work for other loss functions? Can we predict the sharp dip in performance as we increase the fraction of pruned examples? Do rankings computed for one architecture work well for pruning the dataset for a different architecture?

## Significance

On one hand, I think the findings are unexpected and can make follow-up work ask many interesting questions. On the other hand, I don't see a practical use case for compressing the dataset this way. Namely, there are quite a few hyper-parameters (EL2N vs GraNd; sliding window location; epoch used for ranking), and I think in the vast majority of cases it will be simpler to just train the network on the whole dataset.

One way to apply this could be to demonstrate that these metrics computed for one architecture (e.g. VGG) work well for other architectures (e.g. ResNet), in which case one could produce an architecture-agnostic ranking of images for each dataset, which would be very useful. If this kind of result could be added (or is already in the paper and I missed it?), I would rate this paper higher in terms of significance.

Alternatively, significance could be improved if the authors used a label-independent metric as I mention in section "Quality", in which case I could imagine applications in semi-supervised / active learning (e.g. one could use these scores to request labels for highly-ranked images).


**Time Spent Reviewing:**

4

---

> ### Author Response · Authors · 2021-08-10
> **Response to reviewer EkuL**
>
> Thank you for your thorough review! Your comments have helped identify ways in which we can make our presentation stronger. We are glad you found our key result very interesting and surprising and hope that elaborating on our motivation/insight/ablations will make a case for advocating for this paper. We will address your comments on
>
> 1. the derivation in Sec 2.2,
> 2. motivation behind the experiments,
> 3. y-independent scores, and
> 4. potential practical use cases.
>
> First, the analysis in Sec 2.2 is correct, though it could be made clearer. To answer your specific questions:
>
> - Lines 114 and 115: $\Delta_t$ is the time derivative of the loss on an arbitrary input $(x^*,y^*)$. Since over time our trajectory depends on the training set $S$, $\Delta_t$ depends on $S$ (or a subset $S_t$ of $S$, which is the actual minibatch seen at time $t$). In other words, the change in loss measured on some $(x^*,y^*)$ is determined by the gradient of the training objective at time $t$, calculated on the minibatch $S_t$.
> - Line 116: We do not see how it contradicts Eq (1) and believe the sign is correct. The time derivative of the weights is the scaled negative loss gradient. However, the gradient of the loss w.r.t. the weights, $g_{t-1}$, should actually be $g_t$. We fixed the typo.
> - Lines 118 and 120: We considered the full training set $S$ instead of a minibatch since the minibatch setting might collapse to a trivial case—if the training example $(x_j, y_j)$ isn't in the minibatch $S_t$, whether we keep it in the training set or not will not affect the minibatch, and therefore the gradient. Thus, it will have no effect on $\Delta_t$. We will clarify this in the paper.
> - Line 124: $(x,y)$ in the lemma is an arbitrary point from the domain, whereas $(x,y)$ in line 124 is a training example. We see how it can be easily missed and confuse the reader. We will change the notation to address this.
> - Lines 131-136: We do not suggest that this happens, but simply describe conditions under which EL2N scores would contain information similar to GraNd scores. Such conditions would be in line with observations in other recent papers (e.g., Fort and Ganguli ‘19, Papyan ‘20, Fort et al ‘20). We will expand on this and add references to the paper.
>
> Next, we clarify the motivation for our experimental design. We have two goals: (1) find computationally tractable methods of finding important examples; (2) investigate how examples of different importance influence the training dynamics. These goals guided our choice of experiments:
>
> Fig 1: Demonstrate GraNd and El2N can be used to distinguish important examples from unimportant ones; it identifies examples (with the lowest scores) that can be dropped from the training set without any loss of test accuracy.
>
> Fig 2: Explore the impact of label noise on these scores, showing that our scores (when they take high values) can also find mislabeled examples; they play a dual role in finding important examples that are not mislabeled as well as noisy/mislabeled examples.
>
> Fig. 3: We demonstrate that these scores provide insight into the effect of important examples on training dynamics: examples with higher scores drive more change in the network’s representation, as measured by kernel velocity. This is an important new result that connects our work to a large body of work examining deep learning through the lens of neural tangent kernels, thereby broadening the appeal of our paper.
>
> Fig 4: We also demonstrate these scores can be used to understand the smoothness of the loss landscape, with higher (lower) scoring examples contributing more roughness (smoothness) to the loss landscape. This is an important new result that connects our work to a large body of work examining deep learning through the lens of loss landscapes, thereby broadening the appeal of our paper.
> We hope this helps and will edit the paper to make it more explicit.
>
> There are of course other interesting questions, many of which you ask in the Clarity section. Some of them we can answer. In the early phase of training, the ranking of examples by EL2N score gets reordered frequently. This however stabilizes quickly. Concretely, for CIFAR10 + ResNet18, the Spearman rank correlation between the EL2N scores computed at epoch 2 and epoch 10 is 0.68 and between epoch 10 and epoch 20 is 0.89. This is also when EL2N scores become effective for data pruning (Figure 1, first row). Why this happens so early in training is an interesting question for future theoretical work. The sharp dip in performance in EL2N scores as we increase the fraction of pruned examples is partially because, as we mention in Sec 4, at smaller budgets the highest score examples seem to be too difficult and hurt generalization. If we exclude some of the highest scores, performance improves; in CIFAR10 + ResNet18, training with 30% of examples with the highest scores achieves ~92% test accuracy (similar to random baseline), but, excluding 2500 of the largest scoring examples achieves ~94% accuracy. There exist methods, such as Area-Under-Margin (https://arxiv.org/abs/2001.10528), for identifying noisy and mislabeled examples at the end of training; doing this early in training is a promising direction for future work.
>
> Regarding ablating the metrics with y-independent baselines: we tried y-independent scores, specifically all the suggested ones as well as $||\partial f_t(x)/\partial w_t - \mathbb{E}_x[\partial f_t(x)/\partial w_t] ||_F$ and scores based on the NTK, before we found our scores which were the most successful ones. In fact, it was while ablating the loss gradient norm that we found, to our surprise, that just the norm of the error vector contained information for getting high accuracy after pruning. We will include these experiments in our appendix as ablations. Also, we confirm that, in all experiments, the 10 random seeds used to compute the score do not intersect with 4 random seeds used to train networks after pruning. We will make this fact explicit in the paper.
>
> However, we disagree with the symmetry argument in part 2 of the Quality section. Our hypothesis is that, even at random initialization, there are images that cluster together in the network's representation space. While which label the random network assigns is random, similar images will get assigned the same label. Under this condition the symmetry argument would not hold. To verify that the natural structure between images and labels is important, we ran a new experiment: for CINIC10 + ResNet18 we calculate the GraNd score at initialization in the same way but using random labels. We then trained networks on data pruned with these scores but with correct labels. The resultant accuracies were significantly worse than random. We will add this experiment to the appendix.
>
> Finally, we elaborate on what we think are practical use cases. Our scores do indeed generalize across convolutional neural networks with different architectures. To test this we calculate EL2N scores for CIFAR10 at epoch 20 with each of three different architectures: ResNet18, ResNet50, and a variant of VGG13 (modified for use with 32x32 images by replacing the final 3 dense layers with one dense layer that predicts logits from the final convolutional representation). The pairwise Spearman rank correlations for these scores are high (>0.87). Then, for pruning levels 40%, 50% and 60% of the data, we train VGG13 and ResNet50 networks on the datasets pruned with ResNet18 scores. We also train each architecture on data pruned using its own score. At all pruning levels and for both VGG13 and ResNet50, the discrepancy in mean test accuracy when trained on data pruned using ResNet18 scores and data pruned using scores from the same architecture is less than 0.0025. Thus, as mentioned in the review, we can use these scores to produce an architecture-agnostic ranking of images for each dataset.
>
> In addition to different networks, EL2N scores calculated using different training runs during hyperparameter tuning also generalize. To test this, we do the following experiment. For CIFAR10 + ResNet18, we run a grid search on the hyperparameters: learning rate (LR) = [0.05, 0.1, 0.2], weight decay = [0.0001, 0.0005, 0.001], and LR decay factor = [0.1, 0.2]. All other hyperparameters are kept the same. This results in 18 training runs, one for each hyperparameter combination. For each run, we calculate the EL2N scores at epoch 20, normalize the scores to [0, 1] and average across runs. Then, for pruning levels 40%, 50% and 60% of the data, we train networks with the original hyperparameters (LR = 0.1, weight decay = 0.0005, LR decay factor = 0.2) on datasets pruned with scores from hyperparameter tuning. We also train networks on data pruned with scores calculated from 10 training runs with the original hyperparameters. The results from these two scores are almost identical: the mean discrepancy in the test accuracy is less than 0.0004. This experiment indicates that we essentially get these scores for free from hyperparameter tuning, a process important for practical applications where the optimal hyperparameters aren’t already known. This has implications for data security: we can only keep the important data points after hyperparameter tuning. Also this result can be used in creative ways to speed up hyperparameter tuning, where the effects of compressing the dataset will add up.
>
> More importantly, the fact that signal for pruning the data exists just 5-10% into training is remarkable and motivates more investigation of the rich early phase of training! Because this score is such a simple metric, it opens up the possibility for both experimental and theoretical work in trying to understand the connection between properties of the data distribution and training dynamics. Thus we think that our work is a valuable addition to the literature.

---

> > ### Comment · Reviewer_EkuL · 2021-08-23
> > **Interesting updates; need further clarification on symmetry argument**
> >
> > Thank you for clarifications and new experiments. In light of the fact that you did ablations with label-independent scores and intend to include it in the paper, and the promising cross-archicteture/hyper-parameter experiments (which I certainly encourage you to grow into a larger-scale study to establish a robust effect), I am looking to increase my score, but I am hesitant due to still not seeing how E2LN and GraNd can be label-dependent at initialization. Specific comments/questions below.
> >
> >
> > > Line 116...
> >
> > I did not suggest changing the sign, but rather shifting indices to $w_{t} - w_{t-1}$. Alternatively, if you change the index of rhs to $g_t$, you should also change dataset index $S_t$, to match Equation (1).
> >
> > > Lines 131-136: We do not suggest that this happens, but simply describe conditions under which EL2N scores would contain information similar to GraNd scores. Such conditions would be in line with observations in other recent papers (e.g., Fort and Ganguli ‘19, Papyan ‘20, Fort et al ‘20). We will expand on this and add references to the paper.
> >
> > Could you please reference specific sections/plots etc to look at?
> >
> >
> > > Regarding ablating the metrics with y-independent baselines: we tried y-independent scores, specifically all the suggested ones as well as  and scores based on the NTK, before we found our scores which were the most successful ones. In fact, it was while ablating the loss gradient norm that we found, to our surprise, that just the norm of the error vector contained information for getting high accuracy after pruning. We will include these experiments in our appendix as ablations. Also, we confirm that, in all experiments, the 10 random seeds used to compute the score do not intersect with 4 random seeds used to train networks after pruning. We will make this fact explicit in the paper.
> >
> > Thank you, this is great to know, and I strongly encourage rigorous ablations with all these metrics in the final revision. However, I am still confused how these scores can be worse than EL2N or GraNd at initialization - please see discussion below.
> >
> > > However, we disagree with the symmetry argument in part 2 of the Quality section. Our hypothesis is that, even at random initialization, there are images that cluster together in the network's representation space. While which label the random network assigns is random, similar images will get assigned the same label. Under this condition the symmetry argument would not hold.
> >
> > Could you please elaborate in detail/math how the symmetry argument does not hold? Precisely, my claim is that for each image and label $(x, y)$ in the training set, EL2N$(x, y)$ and GraNd$(x, y)$ are exactly the same for all possible one-hot $y$ labels at initialization. So at initialization, we might as well define label-independent measures EL2N'$(x)$ := EL2N$(x, y)$ and GraNd'$(x)$ := GraNd$(x, y)$ (for all possible one-hot $y$). So regardless of how images are clustered in the feature space, their GraNd/E2NL scores cannot possibly depend on their ground truth target labels, and hence influence which images will be pruned.
> >
> > > To verify that the natural structure between images and labels is important, we ran a new experiment: for CINIC10 + ResNet18 we calculate the GraNd score at initialization in the same way but using random labels. We then trained networks on data pruned with these scores but with correct labels. The resultant accuracies were significantly worse than random. We will add this experiment to the appendix.
> >
> > Thank you, this is a very intriguing result - could you confirm that in the same setting pruning by GraNd on correct labels leads to a much better result than pruning by GraNd on random labels? Does E2LN have the same tendency to perform poorly when pruned on random labels? In light of the above comment, do you have an explanation for how this could be possible? Perhaps I'm being slow here, but so far I can't see an explanation for this other than random seed noise.
> >
> > > Finally, we elaborate on what we think are practical use cases. Our scores do indeed generalize across convolutional neural networks with different architectures. To test this we calculate EL2N scores for CIFAR10 at epoch 20 with each of three different architectures: ResNet18, ResNet50, and a variant of VGG13 (modified for use with 32x32 images by replacing the final 3 dense layers with one dense layer that predicts logits from the final convolutional representation). The pairwise Spearman rank correlations for these scores are high (>0.87). Then, for pruning levels 40%, 50% and 60% of the data, we train VGG13 and ResNet50 networks on the datasets pruned with ResNet18 scores. We also train each architecture on data pruned using its own score. At all pruning levels and for both VGG13 and ResNet50, the discrepancy in mean test accuracy when trained on data pruned using ResNet18 scores and data pruned using scores from the same architecture is less than 0.0025. Thus, as mentioned in the review, we can use these scores to produce an architecture-agnostic ranking of images for each dataset.
> >
> > Thank you, this is great to hear. To confirm, in this case all EL2N-pruned datasets did significantly better than random subsets? Do you have an idea whether this tendency generalizes to GraNd, other score computation epochs? Just want to get a sense whether this is a general trend or a specific setting.
> >
> > > In addition to different networks, EL2N scores calculated using different training runs during hyperparameter tuning also generalize...
> >
> > Thank you, this is also an exciting result!

---

> > > ### Author Response · Authors · 2021-08-29
> > > **Symmetry argument and response to other comments**
> > >
> > > Thank you for your comments. We are glad that you found our clarifications and new experiments compelling.
> > >
> > > > promising cross-architecture/hyper-parameter experiments (which I certainly  encourage you to grow into a larger-scale study to establish a robust  effect)
> > >
> > > We have added a **Robustness of data pruning across architectures and hyperparameters** section to the appendix where we present the experiments quoted in our previous response. We are also in the process of running experiments on other dataset and architecture combinations.
> > >
> > > **Line 116:** You are right, both indices need to be shifted to $g_t$ and $S_t$ to match Equation (1). Thank you for noticing this, we have corrected the error.
> > >
> > > **Lines 131-136:** Here are the referenced sections:
> > >
> > > - Fort and Ganguli ‘19, Section 5.1 discusses the orthogonality of mean logit gradients, and logit gradients clustering around the mean.
> > > - Fort et al ‘20, Figures 12D-14D provide evidence that mean logit gradient directions evolve rapidly early in training and then stabilize (as measured by cosine distance).
> > >
> > > >  I strongly encourage rigorous ablations with all these metrics in the final revision
> > >
> > > We have added a **Label-Independent Scores Ablations** section to the appendix with experiments comparing our scores to label-independent versions. Here we presented the experiment with randomized labels that we mentioned in our previous response.
> > >
> > > > Could you please elaborate in detail/math how the symmetry argument does not hold?
> > >
> > > Sorry, we misunderstood what you meant by symmetry the first time. Your follow-up question has clarified your observation to us.
> > >
> > > Indeed, you are correct that, at initialization, the formal GraNd and EL2N scores (defined in term of expectations over all possible initializations) are label independent for the reasons you state. The explanation for why the scores appear to be label dependent at initialization has to do with the fact that we approximate the expectation with a finite number of initialized networks.
> > >
> > > In more detail, a randomly initialized network does not produce a uniform distribution over labels, although, as your symmetry argument recognizes, the expected distribution is uniform. The effect of averaging over 16 networks is not strong enough to obtain a uniform distribution. What we find empirically is that certain labels are assigned higher probability at initialization and that this preference holds across the entire data set. As we go from 16 networks to 200 to 1024, we see these preferences (“noise”) vanish and label independence arises. When only 16 networks are used, the average preferences are far from uniform, and so, when labels are then randomized, the variance due to using a moderate number of networks is amplified.
> > >
> > > On the other hand, we observe high correlation (spearman rank coeff. 0.85) when computing the GraND score on two different sets of 10 initializations, suggesting that the signal we take advantage of at initialization is not just due to random seeds but is consistent among initializations.
> > >
> > > > could you confirm that in the same setting pruning by GraNd on correct labels leads to a much better result than pruning by GraNd on random labels?
> > >
> > > Yes, this is the case.
> > >
> > > > Does E2LN have the same tendency to perform poorly when pruned on random labels?
> > >
> > > We have not tried the same experiment with EL2N scores at initialization. The reason is that, for difficult tasks (CINIC10 and CIFAR100), pruning using EL2N at initialization does not do significantly better than random. We observe that the robust effect of EL2N only comes in after some training. Therefore we didn’t explore further whether label randomization impairs EL2N performance at init, which is already close to random for difficult tasks.
> > >
> > > > To confirm, in this case all EL2N-pruned datasets did significantly better than random subsets?
> > >
> > > Yes.
> > >
> > > > Do you have an idea whether this tendency generalizes to GraNd, other score computation epochs?
> > >
> > > This trend generalizes to all EL2N score calculation epochs at which pruning by EL2N scores obtains performance better than pruning a random subset. In our Figure 1 experiments, this happens for all epochs starting at epoch 4. We have not tried the earlier epochs because EL2N's performance is not robust.   In our new **Robustness of data pruning across architectures and hyperparameters** section we show that this trend holds across datasets and score calculation epochs. We have not tried cross-architecture with GraNd scores yet but we intend to also run experiments and add them to this Appendix section.

---

> > > > ### Comment · Reviewer_EkuL · 2021-08-31
> > > > **Still confused about symmetry**
> > > >
> > > > Thank you for your replies.
> > > >
> > > > > The effect of averaging over 16 networks is not strong enough to obtain a uniform distribution. What we find empirically is that certain labels are assigned higher probability at initialization and that this preference holds across the entire data set. As we go from 16 networks to 200 to 1024, we see these preferences (“noise”) vanish and label independence arises.
> > > >
> > > > While it is certainly possible that, averaged over some 16 network initializations, higher probabilities are assigned to some class $C_1$, this class is random, and if you average other another 16 random seeds, the higher probability will be assigned to another random class $C_2$, correct? Therefore I don't see how this preference could be useful anywhere, as it is indeed just noise.
> > > >
> > > > > When only 16 networks are used, the average preferences are far from uniform, and so, when labels are then randomized, the variance due to using a moderate number of networks is amplified.
> > > >
> > > > I don't quite understand what this means, could you please explain precisely / mathematically what is amplified compared to natural image distribution, and why exactly does it make pruning work worse?
> > > >
> > > > > On the other hand, we observe high correlation (spearman rank coeff. 0.85) when computing the GraND score on two different sets of 10 initializations, suggesting that the signal we take advantage of at initialization is not just due to random seeds but is consistent among initializations.
> > > >
> > > > There is a signal from the input $x$, so it is expected that scores will be correlated. To validate your claim that there is signal in $y$, the number $0.85$ should be shown to be higher than if you were to repeat the same experiment but with labels randomized in one set.
> > > >
> > > > >> could you confirm that in the same setting pruning by GraNd on correct labels leads to a much better result than pruning by GraNd on random labels?
> > > >
> > > > > Yes, this is the case.
> > > >
> > > > In light of the above, I am still quite puzzled by this result. Do I also understand correctly that per your argument above, if you were to estimate these GraNd scores without shuffling labels, on many more random inits (e.g. 1024), then it would also do just as bad as if you were to shuffle labels, and much worse than when you average over 16 inits?

---

> > > > > ### Author Response · Authors · 2021-09-02
> > > > > **Popping up a level**
> > > > >
> > > > > Before we dive into the details, we wanted to pop up a level and put this thread into context. If we understand correctly, we've addressed essentially almost all of your concerns. We're now discussing one very particular effect: the role of labels in the GRanD score computed at initialization (most of our paper focuses on pruning using information later in training, where performance is robustly better across both architectures and datasets, unlike pruning using GraND scores at init).  To be clear, this only remaining concern is but one very small aspect of our paper. It would be very helpful to know if you still harbor concerns serious enough to recommend rejection, so that we can focus on those concerns given the limited time. We do believe we are already at a point where we have addressed your most pressing concerns.
> > > > >
> > > > > Now, turning to this one aspect of our paper: the role of labels in the GRanD score at init. You have raised 3 questions:
> > > > >
> > > > > 1. Why is the GRanD score label dependent? We have answered this: namely while the formal GRanD score averaged over all possible inits is label independent at initialization, the GRanD score computed by averaging over a finite number of inits can acquire label dependence.
> > > > > 2. Why does pruning with the GRanD score at init do slightly better than pruning at random?
> > > > > 3. Why does randomizing the labels of the training data then pruning with GRanD score with random labels do slightly worse than pruning randomly?
> > > > >
> > > > > We believe points 2 and 3 are very interesting observations, but to put them in context, it is worth discussing quantitatively effect sizes.  For example, we find on CINIC10+ResNet18 when we prune 30 percent of the data, we obtain:
> > > > >
> > > > >
> > > > > 1. 89% baseline accuracy with random pruning
> > > > > 2. 89.5% accuracy with GraND score at init (averaging over a small number of inits)
> > > > > 3. 87% accuracy with GranND score computed on randomized labels at init (averaging over the same number of inits)
> > > > >
> > > > > Points 1 and 2 above can be found in our Figure 1 middle bottom panel, and Point 3 above was computed during this rebuttal period.  While these are statistically significant differences, on an absolute scale, these effect sizes are not large. And in any case, pruning by GRanD score at init is not the best algorithm (i.e. pruning by El2N at epoch 20 achieves greater than 90% accuracy).
> > > > >
> > > > > We have been digging into the role of labels, and label randomization in GRaND, but, honestly, we do not believe we can develop a complete mathematical theory of how the GRaND score behaves with respect to labels when averaging over a small number of inits.  But we also believe it would be overly harsh to reject this paper based on this one minor aspect of our paper alone.  This is fundamentally an empirical paper that yields extremely interesting observations that could form the fodder for future theory.  There is great precedent for such papers in our field - papers which revealed interesting empirical phenomena with no theoretical explanation whatsoever  (some notable examples include lottery tickets, adversarial examples, and rethinking generalization).  We appreciate your help in teasing apart what's going on in this particular ablation, but we also think it would be unfortunate to delay the publication of this work, unless there were other major concerns. We are very happy to write our discussion in such a way as to bring this point out as a clear point for future research.
> > > > >
> > > > > Nevertheless, in the absence of a complete mathematical theory explaining the above questions, we can say a few things based on recent new experiments done during the rebuttal period.
> > > > >
> > > > > First we have noted that each randomly initialized network assigns non-uniform probabilities for any fixed input x (even though, in expectation over initializations, the label probabilities are uniform). Thus when averaging over a small number of inits,  the averaged label probabilities are still non-random. Similarly, the mean logit gradient does not depend on the label, but for a fixed network, there are differences and these don't get completely averaged away with only 16 networks (or even with ~1000 networks, as indicated by experiments we've run in the past few days). Note that we are NOT arguing that this noise is *always beneficial*. (We apologize if we inadvertently conveyed that suggestion in our previous response.) We agree with your instinct that these "preferences" are likely not universally useful across all datasets and architectures.
> > > > >
> > > > > We've been visualizing the GraNd scores at initialization and see some interesting patterns. E.g., as we've mentioned, there is a consistent bias over label probabilities across the entire data set (every input has a similar pattern of probability assigned to class labels, across the set of class labels).  Thus it *may* be the case that if there is a particular correlation between the location of an input and its correct class label, then computing a GRaND score with the correct class labels versus randomized class labels could yield different performances (slightly higher or lower than random), when averaging over a small number of inits.  Whether it is actually higher or lower than random with either correct or randomized labels likely depends on the subtle nature of correlations between input locations and correct class labels that we do not at the moment understand.
> > > > >
> > > > > Now if we increase the number of inits we average over, of course the GRaND score will eventually become label independent.  Then we expect that the performance attained by pruning via GRaND score with the correct labels, and pruning via GRaND score with randomized labels,  will be similar to each other.  Whether or not this similar performance is better or worse than pruning via random would then depend on the structure of the input distribution over x alone and not on the labels.  One cannot jump to the conclusion, raised by the reviewer that: “i​​f you were to estimate these GraNd scores without shuffling labels, on many more random inits (e.g. 1024), then it would also do just as bad as if you were to shuffle labels, and much worse than when you average over 16 inits?”   It could alternatively be the case that shuffling labels with a small number of inits does more harm than not shuffling labels and using a larger number of inits. While we have not precisely pinpointed the way that randomizing the labels and pruning via the GRaND score harms performance, we do believe that it does harm performance more than pruning via GRaND score with the correct labels, and increasing the number of inits, based on our experiments so far.
> > > > >
> > > > > It is worth highlighting that we are not recommending using GraNd scores at initialization if you randomize the labels. Moreover, using GraND scores at init without randomizing labels is not even our best algorithm across dataset-architecture combinations. Indeed, it is but a small part of our paper. We actually recommend training for a small amount of time then computing scores - this yields more robust performance across all dataset/architecture combinations.
> > > > >
> > > > > Again, we believe these are interesting questions, and we will clearly write our discussion to bring these questions to the fore as interesting directions for future research.  In writing our discussion this way, we believe this paper will garner even more attention and achieve more impact by spurring new research.
> > > > >
> > > > > We hope now that we believe we have addressed your concerns, you will raise your score. We are happy to discuss further also as needed.

---

> > > > > > ### Comment · Reviewer_EkuL · 2021-09-03
> > > > > > **Raising score due to architecture/hparam robustness, and in expectation of other changes; interpretation/ablations remain a weakness**
> > > > > >
> > > > > > Thank you for your reply. I appreciate the effort, and I went through the paper one more time. Overall I remain unconvinced by discussion about GraNd and find it to be a significant weakness of the paper. However, I will raise my score since you've shown your method to be robust to hyperparameters and architectures, which I find to be useful.
> > > > > >
> > > > > > > Before we dive into the details, we wanted to pop up a level and put this thread into context. If we understand correctly, we've addressed essentially almost all of your concerns. We're now discussing one very particular effect: the role of labels in the GRanD score computed at initialization (most of our paper focuses on pruning using information later in training, where performance is robustly better across both architectures and datasets, unlike pruning using GraND scores at init). To be clear, this only remaining concern is but one very small aspect of our paper. It would be very helpful to know if you still harbor concerns serious enough to recommend rejection, so that we can focus on those concerns given the limited time. We do believe we are already at a point where we have addressed your most pressing concerns.
> > > > > >
> > > > > > I disagree that this is a minor point. Figure 1 shows GraNd at init to be very competitive, and this merit is highlighted several times in the paper. In addition, GraNd in general is central to the paper and EL2N is motivated as an approximation.
> > > > > >
> > > > > > > Now, turning to this one aspect of our paper: the role of labels in the GRanD score at init. You have raised 3 questions:
> > > > > >
> > > > > > > Why is the GRanD score label dependent? We have answered this: namely while the formal GRanD score averaged over all possible inits is label independent at initialization, the GRanD score computed by averaging over a finite number of inits can acquire label dependence.
> > > > > > Why does pruning with the GRanD score at init do slightly better than pruning at random?
> > > > > >
> > > > > > I disagree with characterizing this as "slightly" - Figure 1 CIFAR-10 shows GranD at init doing pretty much as good as possible. On CINIC-10 and CIFAR-100 it's about halfway between random baseline and training on the whole set (and more than halfway between baseline and the best pruned result). This is not a slight effect.
> > > > > >
> > > > > > > Why does randomizing the labels of the training data then pruning with GRanD score with random labels do slightly worse than pruning randomly?
> > > > > > We believe points 2 and 3 are very interesting observations, but to put them in context, it is worth discussing quantitatively effect sizes. For example, we find on CINIC10+ResNet18 when we prune 30 percent of the data, we obtain:
> > > > > >
> > > > > > > * 89% baseline accuracy with random pruning
> > > > > > > * 89.5% accuracy with GraND score at init (averaging over a small number of inits)
> > > > > > > * 87% accuracy with GranND score computed on randomized labels at init (averaging over the same number of inits)
> > > > > >
> > > > > > > Points 1 and 2 above can be found in our Figure 1 middle bottom panel, and Point 3 above was computed during this rebuttal period. While these are statistically significant differences, on an absolute scale, these effect sizes are not large. And in any case, pruning by GRanD score at init is not the best algorithm (i.e. pruning by El2N at epoch 20 achieves greater than 90% accuracy).
> > > > > >
> > > > > > Again, I disagree that this is "slightly worse" (you yourself called it "significalty worse" in your earlier messages) - the 2% drop on CINIC-10 when you randomize the labels is 4X larger than the 0.5% gain when you don't, and 2X larger than the 1% gain with your best method.
> > > > > >
> > > > > > > We have been digging into the role of labels, and label randomization in GRaND, ...
> > > > > >
> > > > > > > Now if we increase the number of inits we average over, of course the GRaND score will eventually become label independent. Then we expect that the performance attained by pruning via GRaND score with the correct labels, and pruning via GRaND score with randomized labels, will be similar to each other. Whether or not this similar performance is better or worse than pruning via random would then depend on the structure of the input distribution over x alone and not on the labels. One cannot jump to the conclusion, raised by the reviewer that: “i​​f you were to estimate these GraNd scores without shuffling labels, on many more random inits (e.g. 1024), then it would also do just as bad as if you were to shuffle labels, and much worse than when you average over 16 inits?” It could alternatively be the case that shuffling labels with a small number of inits does more harm than not shuffling labels and using a larger number of inits. While we have not precisely pinpointed the way that randomizing the labels and pruning via the GRaND score harms performance, we do believe that it does harm performance more than pruning via GRaND score with the correct labels, and increasing the number of inits, based on our experiments so far.
> > > > > >
> > > > > > Thank you for conducting all these experiments in such a short time. Unfortunately I still find the data on this confusing enough to suggest further investigation. To recap my concern:
> > > > > >
> > > > > > 1. Empirical GraNd at init is a strong baseline for pruning, frequently outperforming pruning at later epochs, depending on various settings. This makes it, in my opinion, important to understand.
> > > > > > 2. Theoretical GraNd at init as defined is label-independent; however, as your label-randomized experiment has shown, empirical GraNd is highly label-reliant, i.e. it needs the correct labels to provide the pruning benefit. Theoretical GraNd would be agnostic to randomizing, but you report a 2.5% absolute drop on CINIC-10, which is, as I mentioned above, a very strong effect relative to other results in the paper.
> > > > > > 3. Due to the magnitude of the effect above, I conclude that empirical GraNd is very different from theoretical GraNd. It could be that theoretical GraNd, if it were evaluated, gives better or worse performance, but its mechanism must be different from empirical GraNd.
> > > > > >
> > > > > > Therefore what is being evaluated in experiments and giving the pruning benefits (empirical) is clearly very different from what is presented as definition and motivation (theoretical). Further, apparently the number of evaluations is an important and somewhat mysterious hyper-parameter for empirical GraNd, and it's given very little attention in the paper (per discussion above, current GraNd behaves very differently from how it would if it were evaluated many more times; at the same time the paper says that averaging over multiple initializations is important, and single sample evaluations don't do well). The impact of this hyperparameter should be rigorously studied in the next revision.
> > > > > >
> > > > > > > It is worth highlighting that we are not recommending using GraNd scores at initialization if you randomize the labels. Moreover, using GraND scores at init without randomizing labels is not even our best algorithm across dataset-architecture combinations. Indeed, it is but a small part of our paper. We actually recommend training for a small amount of time then computing scores - this yields more robust performance across all dataset/architecture combinations.
> > > > > >
> > > > > > See my replies above - I find GraNd at init to be quite competitive and this is highlighted in the paper.
> > > > > >
> > > > > > > Again, we believe these are interesting questions, and we will clearly write our discussion to bring these questions to the fore as interesting directions for future research. In writing our discussion this way, we believe this paper will garner even more attention and achieve more impact by spurring new research.
> > > > > >
> > > > > > > We hope now that we believe we have addressed your concerns, you will raise your score. We are happy to discuss further also as needed.
> > > > > >
> > > > > > I will raise my score due to good results on architecture/hyperparameter robustness, and in expectation of other changes (ablation against label-independent metrics + impact of number of evaluations on performance of GraNd/EL2N + rework presentation / theory based on the mismatch between empirical and theoretical GraNd). Thank you for your active participation.

---

### Official Review · Reviewer_qhFJ · 2021-07-18

**Rating:** 6
**Confidence:** 4

**Summary:**

The authors propose GraNd score based on expected gradient loss to filter out low impact examples from the training. The proceed to empirically show that pruning examples with low GraNd score is competitive with state of the art methods with the benefit of being able to use the score early at training (as to save on compute).

The paper then discusses the relationship of GraNd with EL2N---the norm of the prediction vector minus the one hot of the true label, showing that EL2N can be used to improve the performance of models by removing the examples with high EL2N.

Then, the authors go and compare different scoring ranges for pruning to find that the highest scores are not beneficial when there is label noise. Moreover, they experimentally show that high scoring examples drive learning velocities in NTK and contribute to rougher loss landscape.



**Limitations And Societal Impact:**

yes

**Main Review:**

While the score the authors suggest is interesting I have some reservations about the paper. First, the 'theoretical derivation' is very unclear (I mention a couple of examples below) and also I'm not completely sure whether it is correct. Therefore, this score could be just a random functional that might or might not be too interesting.

Moreover, the results are a bit scattered and I'm not sure what is the main message of the paper. Do we want to find important examples? If so, I would first try to define what is an important example. Then for every experiment I would want to understand why this experiment was conducted, and why this demonstrates that the metric indeed finds examples that are important. To me it seems that the main experiment is the pruning one and it seems that the metric proposed by this paper is slightly worse than the forgetting score so the contribution is incremental. (Also, you should compare to Area-Under-Margin, see comment below).

Other comments:

* When the authors discuss time derivatives it is unclear what they mean. Unless doing gradient flow (which was not mentioned in the paper) time works in discrete steps, so I don't see why the time derivative is well defined. If the point was to discuss discrete derivatives it is unclear why the equation below eq 2 has \approx because then it should be =.

* In Lemma 2.2. I'm not sure why \|d\ell(f(x,y))  / dw_t \| is finite and why does it not depend on (x,y) as mentioned in the discussion above? For the former, there could be example for which the loss is unbounded after some step, for the later, to me it seems that the derivative of the loss at example (x,y) has to be dependent on (x,y). Also, would appreciate an explanation to what does the chain rule mean (and why it holds) when the derivative discrete.

* Not sure what 'maintaining \Delta_t x` means in line 128.

* I assume that in equation 4 the expectation is over initialization, this should be mentioned.

* In figure 1. I would remove 'for pruning strategy see left' and have a large legend that spans the entire figure so it is clear that the legend is relevant to everything.

* while forget score is calculated later in training it seems to be beating both proposed measures in the paper.

* I think this paper is relevant and probably should be cited and compared against: https://arxiv.org/abs/2001.10528

+++++
Post rebuttal:
I would like to thank the authors for their detailed review and alleviating some of my concerns and decided to change the score 5 -> 6. I think some minor edits along the lines of the rebuttal could help the paper's readability and motivation.

**Time Spent Reviewing:**

6 hours

---

> ### Author Response · Authors · 2021-08-10
> **Response to reviewer qhFJ part 1/2**
>
> Thank you for your thorough review! Your questions have helped clarify the main messages of our paper. We address your main concerns about:
>
> 1. the definition of example importance,
> 2. the core message and experimental motivation,
> 3. our theoretical analysis, and
> 4. comparisons to both Area-Under-Margin and forget scores.
>
> First and foremost, you raise the question, **what is an important training example**? While our theoretical analysis is focused on the “importance”/impact of an individual example on training dynamics, there is, of course, complex dependence between examples and how they affect training. A useful notion that we use ourselves, and which we will add to the paper, is the idea of a SUBSET of training examples being “epsilon-good”, which means that training on this subset leads us to training error (or test error, depending on the situation we’re focusing on) being no worse than epsilon off from that of a run of SGD on the full data set. Our approach is then to search for large epsilon-good sets by ranking examples according to a score and thresholding. There are, of course, other approaches to this problem, including a wide array of coreset methods. The challenge in deep learning is the scale of data, model, and the cost of retraining (which most coreset methods must repeatedly invoke). Remarkably, a metric that ranks the “importance” of individual examples seems to allow us to find epsilon-good sets for epsilon ~= 0, whose size is a large constant fraction (near 50%). Regarding the definition of “importance of an example”: the actual rigorous notion is that of an epsilon-good set. The idea of an “important” example, as defined implicitly by our score, was our way of approaching the problem of ranking examples. For large data sets we cannot even hope to consider all pairs of data. So we consider measuring the impact/importance etc. of an example by measuring the effect of removing this one example from training and we consider only the impact over one step, which we further approximate by assuming the step size is very small (i.e., gradient flow). Regardless of how we got there, it is remarkable to us that the behavior of gradient flow at initialization provides a signal useful for finding large epsilon-good sets.
>
> The goal of understanding epsilon-good sets (by way of searching for them) and doing so by attempting to measure the “importance”/impact of individual example on training are the **core goals** of our paper. Our work connecting training dynamics (linear mode connectivity) to our measures of importance are towards this goal of understanding. Lines 43-50 lay out our goals more broadly, but we can expand/clarify these along the lines outlined above. These core goals motivate both our theoretical analysis and our experiments as we explain next.
>
> Next, we argue that our **theoretical analysis** is sound, and is motivated by the goals outlined above. The logic that brings us to the GraNd score appears in Sec 2.2 in words. The GraNd score is designed to approximate and upper bound the impact of the single training example. (Formally, it is measuring the epsilon-goodness of a set of size n-1.) However, approximations are necessary. First, instead of training to the end, we perform a one step analysis of the training loss after training on all the data and following one gradient step, vs. the training loss after training on all of the data except for a single training example. We show that the difference in these two losses can be upper bounded by the gradient norm of the example that was dropped. The El2N score is a further approximation of the GraNd score that only keeps error information at the output.
>
> Thus our scores are theoretically well motivated for the problem of measuring the impact of a single example at one particular point in training. Remarkably, the ranking they induce allows us to find large epsilon-good sets. This is, at the moment, something that we are probing empirically. We show that using these strikingly simple scores as proxies for example importance, we can remove more examples without sacrificing accuracy compared to randomly selecting examples. Thus while we appreciate the reviewer’s concern that these scores could be “just a random functional that might or might not be too interesting,” we believe their combined theoretical motivation and empirical outperformance of a random pruning baseline rules out this possibility. Since you found our scores interesting but that excitement was undercut by concerns around our analysis, we hope that clarifying the framing will rekindle your interest. We also respond to your specific comments on Sec 2.2 at the end of part 2 of this response.

---

> > ### Author Response · Authors · 2021-08-10
> > **Response to reviewer qhFJ part 2/2**
> >
> > With the clarification of example importance and the theoretical motivation of our scores in hand, we next clarify the **motivation for each of our experiments**. These motivations were all in the paper, but we hope this clarification helps. We will edit the paper to make the motivation for each experiment more explicit in each section.
> >
> > The data pruning experiments, in which we remove examples sorted by our scores and then measure final test accuracy when training on the remaining examples take center stage because they provide strong evidence that our computationally tractable scores yield a good, and even practically useful ranking of examples to find epsilon-good sets. In essence, if our score finds examples that can be dropped from the training set without losing test accuracy, then our score (or small values of it) yields a practical method for finding and removing unimportant examples, and training only on important ones to yield an epsilon-good set. As an additional benefit, we are able to use our score to prune non-trivial fractions of the dataset early in training, even for completely new initializations and architectures. This opens our method up to creative practical applications. Our second goal of investigating how examples of different importance influence training dynamics motivates the rest of our experiments. In summary, our experiments and their motivation are as follows:
> >
> > Fig 1: Demonstrate GraNd and El2N can be used to find important examples very early in training.
> >
> > Fig 2: Explore the impact of label noise on these scores, showing that our scores (when they take high values) can also find mislabelled examples - so they play a dual role in finding important examples that are not mislabelled as well as noisy/mislabelled examples.
> >
> > Fig. 3: We demonstrate that these scores provide insight into the effect of important examples on training dynamics: examples with higher scores drive more change in the network’s representation, as measured by kernel velocity. This provides an important new result that connects our work to a large body of work examining deep learning through the lens of neural tangent kernels, thereby broadening the appeal of our paper.
> >
> > Fig 4: We also demonstrate these scores can be used to understand the smoothness of the loss landscape, with higher (lower) scoring examples contributing more roughness (smoothness) to the loss landscape. This provides an important new result that connects our work to a large body of work examining deep learning through the lens of loss landscapes, thereby broadening the appeal of our paper.
> > Thank you for your comment on motivating the experiments better, clarifying this will make our paper stronger.
> >
> > Finally, we compare our paper to other work. Thank you for bringing our attention to Area-Under-Margin (AUM); we will discuss the connections to our work in the related work section. We find two key differences: first, AUM is calculated by tracking a statistic through training while we focus on instantaneous information in the early phase of training. Second, AUM is specifically aimed at identifying mislabeled points and is especially effective on noisy datasets. Our approach, on the other hand, ranks points by importance and therefore also prunes redundant/easy points. These methods are orthogonal and can be used together for even stronger results. Next, though pruning by forget scores calculated at *the end of training* does achieve equivalent or better accuracy than our method, it is not a practical pruning method if we wish to avoid training on all the data in the first place. Additionally, the authors of the forgetting paper didn't directly evaluate pruning based on forget scores computed early in training. That this works so well is an important finding that, to our knowledge, hasn't been reported yet. However, even if we restrict to the practical setting of pruning based on scores computed early in training, our method performs as well as or better than forget scores calculated early in training (Figure 1, first row). More importantly, it is quite surprising that a simple instantaneous score like EL2N *just 5-10% into training* provides a signal containing almost as much information as forget scores which are calculated over the entire training run! This observation motivates future experimental and theoretical work examining the rich early phase of training and the influence of the data distribution on training dynamics. We think that this is crucial to understanding the remarkable success of deep learning, thus making our work a valuable addition to the literature.
> >
> > We now address the list of other comments, several of which relate to concerns about our analysis in Sec 2.2.
> > - The time derivatives arise because, as stated in Lines 111-112, “In order to simplify our analysis, we approximate the training dynamics as if they were in continuous time.” The score we use is literally the instantaneous time derivative of gradient flow. Therefore, this aspect of gradient flow at initialization appears to provide us with a signal to prune a significant amount of data. It is surprising that a property of the initialization determines long-run aspects of the nonconvex optimization involved in training.
> > - The gradient of the loss does depend on $(x,y)$, as suggested by our notation. Perhaps the reviewer is referring to $c$ in line 123. We see how it may be confusing: in the lemma we index the training points by $j$, but drop $j$ in the discussion paragraph below the lemma. $c$ only depends on an arbitrary point that we measure the change in loss on, and not on the training points which we are comparing (as stated in the lemma, “for all $(x,y)$, there exists $c$...”). Thus all the statements are correct. Further, if the gradients are bounded, which we believe is a reasonable assumption along the trajectory, and the domain $X\times Y$ is bounded, $c$ can be chosen to be the sup of the gradients over the domain. None of our conclusions would change. We are happy to redefine $c$ to be the sup if it would improve the presentation.
> > - “Maintaining $\Delta_t(x)$” - preserving may be a better word.
> > - Since $\chi_t$ is a function of $(x,y)$, thus making $(x,y)$ non-random, the expectation is over the remaining randomness—the weights at time t (which depend on a random initialization, random minibatch sequence, GPU noise, etc.)
> > - Thank you for suggesting figure edits to improve presentation.

---

> > > ### Author Response · Authors · 2021-08-31
> > > **Remaining concerns**
> > >
> > > We only just noticed that you updated your review with a "post rebuttal" comment. (OpenReview didn't announce this by email unfortunately.) We're happy that we were able to alleviate some of your concerns, enough to convince you to increase your score. Your comment suggests that "minor edits along the lines of the rebuttal could help the paper's readability and motivation". We are of course going to take every drop of insight from this thread and use it to improve our paper. Since you're suggesting minor edits, we'd like to ask you to consider recommending Accept. At NeurIPS, Weak Accepts generally aren't accepted, which doesn't seem to be your intention, but it will be the effect. If there is a substantial issue that is preventing you from recommending a straight Accept, perhaps you can raise that with us, so that we can attempt to address it.

---

### Official Review · Reviewer_3e6T · 2021-07-18

**Rating:** 8
**Confidence:** 4

**Summary:**

This paper proposes a heuristic to prune the dataset for training by using the norm of the gradient for an ensemble of random networks as a proxy score. They provide some intuition for the proposed method and conduct experiments to verify that the method works well in practice.

**Limitations And Societal Impact:**

Yes, the authors have discussed these aspects well

**Main Review:**

Strengths:
- The problem is well-motivated with clear potential applications to speed up training and save computational costs
- The method proposed is sound. It also poses an interesting question about how the initial random network 'essentially' determines characteristics of future training. The connection to NTK and forgetting score is also interesting.
- The experimental set up is reasonable, and the empirical results suggest the method works well in practice

**Time Spent Reviewing:**

2 hrs

---

> ### Author Response · Authors · 2021-08-10
> **Response to reviewer 3e6T**
>
> Thank you for your very positive review. We hope you can advocate for this paper with the other reviewers, as you see clear potential applications and empirical evidence of a method that works well in practice. You have nicely summarized one of our important findings: some key aspects of training are seemingly fixed at initialization. We are glad you agree that this is a nontrivial observation and that our paper should have substantial impact, not only in discovering practical algorithms that work well, but also in uncovering striking phenomena that open up new challenges and opportunities in understanding deep learning for experimentalists and theorists alike.

---

### Decision · Program_Chairs · 2021-09-27

**Decision:**

Accept (Poster)

**Comment:**

This paper proposes a new heuristic to prune the dataset for training a neural network by using the norm of the gradient for an ensemble of random networks as a proxy score (error norm: EL2N, gradient norm: GraNd).  Authors show that pruning examples with low GraNd score is competitive with state of the art methods with the benefit of being able to use the score early in the training. Using these metrics, they shed light on how the underlying data distribution shapes the training dynamics: they rank examples based on their importance for generalization, detect noisy examples and identify subspaces of the model’s data representation that are relatively stable over training.


Common weaknesses pointed out were some lack of clarity (`ghFJ`, `EkuL`), unclear practical value (`V5c3`, `EkuL` ) and efficacy(`V5c3`). The authors actively engaged in the discussion and provided explanations and results to address the reviewer's concerns. The AC believes analyses performed and explanations given during the discussion would improve the final version of the paper further and strongly encourages the authors to adopt them. Questions regarding architecture / hyperparameter robustness and lack of ablations, raised by reviewer `EkuL`, were also resolved.

After the discussion, all reviewers recommended acceptance (1 clear accept, 3 weak accepts). While efficacy and practical applications are not fully developed, for this type of theory inspired work those should be criteria for higher distinction than acceptance. The AC believes strengths outweigh the weaknesses. The problem studied in the paper is well motivated and provides interesting insights for understanding deep neural networks and, as authors claim, have potential to inspire new methods and new theories of deep learning. The AC recommends acceptance.